# Malignant Pleural Mesothelioma: Genetic and Microenviromental Heterogeneity as an Unexpected Reading Frame and Therapeutic Challenge

**DOI:** 10.3390/cancers12051186

**Published:** 2020-05-07

**Authors:** David Michael Abbott, Chandra Bortolotto, Silvia Benvenuti, Andrea Lancia, Andrea Riccardo Filippi, Giulia Maria Stella

**Affiliations:** 1Department of Medical Sciences and Infective Diseases, Unit of Respiratory Diseases, IRCCS Policlinico San Matteo Foundation and University of Pavia Medical School, 27100 Pavia, Italy; david.abbott01@universitadipavia.it; 2Unit of Radiology, Department of Intensive Medicine, IRCCS Policlinico San Matteo Foundation and University of Pavia Medical School, 27100 Pavia, Italy; c.bortolotto@smatteo.pv.it; 3Candiolo Cancer Institute, FPO—IRCCS—Str. Prov.le 142, km. 3,95—10060 Candiolo (TO), Italy; silvia.benvenuti@ircc.it; 4Unit of Radiation Therapy, Department of Medical Sciences and Infective Diseases, IRCCS Policlinico San Matteo Foundation and University of Pavia Medical School, 27100 Pavia, Italy; a.lancia@smatteo.pv.it (A.L.); a.filippi@smatteo.pv.it (A.R.F.)

**Keywords:** asbestos, genetics, inflammation

## Abstract

Mesothelioma is a malignancy of serosal membranes including the peritoneum, pleura, pericardium and the tunica vaginalis of the testes. Malignant mesothelioma (MM) is a rare disease with a global incidence in countries like Italy of about 1.15 per 100,000 inhabitants. Malignant Pleural Mesothelioma (MPM) is the most common form of mesothelioma, accounting for approximately 80% of disease. Although rare in the global population, mesothelioma is linked to industrial pollutants and mineral fiber exposure, with approximately 80% of cases linked to asbestos. Due to the persistent asbestos exposure in many countries, a worldwide progressive increase in MPM incidence is expected for the current and coming years. The tumor grows in a loco-regional pattern, spreading from the parietal to the visceral pleura and invading the surrounding structures that induce the clinical picture of pleural effusion, pain and dyspnea. Distant spreading and metastasis are rarely observed, and most patients die from the burden of the primary tumor. Currently, there are no effective treatments for MPM, and the prognosis is invariably poor. Some studies average the prognosis to be roughly one-year after diagnosis. The uniquely poor mutational landscape which characterizes MPM appears to derive from a selective pressure operated by the environment; thus, inflammation and immune response emerge as key players in driving MPM progression and represent promising therapeutic targets. Here we recapitulate current knowledge on MPM with focus on the emerging network between genetic asset and inflammatory microenvironment which characterize the disease as amenable target for novel therapeutic approaches.

## 1. Introduction

Malignant Pleural Mesothelioma (MPM) is an aggressive cancer with very dismal prognosis from diagnosis whose pathogenesis is mainly associated with exposure to asbestos fibers. MPM was first linked to exposure to the industrial fiber asbestos in 1935 [1,2] and it is now well documented that at least 80% cases are caused by exposure to asbestos fibers [1,3]. Due to its extreme versatility, asbestos became very popular in the 1970s and 80s to produce cements, tiles, yarn, toys, jewelry, pipe lining and more [3,4]. In addition, its temperature-resistant properties made it particularly appealing for the insulation and heating trades [3,5]. Unfortunately, these resistance properties have rendered the disposal of the carcinogenic asbestos-laden materials nearly impossible (fittingly, the word asbestos comes from the Greek word for “inextinguishable”), posing formidable epidemiological challenges [5]. MPM is still lacking effective therapies and median survival is around 13–15 months after diagnosis [5]. Growing evidence suggests that asbestos-induced damage is associated to the generation of an inflammatory microenvironment that may support tumor growth, possibly in association to genetic predisposition [6,7,8]. Here we describe current knowledge on MPM, mainly focusing on the heterogeneous context and complex crosstalk between genetics and microenvironment that are emerging as driving force for tumor progression. A deeper understanding of these processes may allow a proper patients stratification as well as the identification of novel and effective therapeutic approaches.

## 2. Epidemiology and Causative Agents

Though mesothelioma is rare and the production of materials with asbestos has been illegal for more than 20 years in many counties—although it is not banned in some others—the incidence of MPM is still rising. This is mainly due to the 20 to 40-year latency of the asbestos effects in an aging, genetically susceptible population. Since 1994 the World Health Organization (WHO) has been tracking epidemiologic data for Malignant Pleural Mesothelioma (MPM) [9]. From 1994 to 2008 the WHO mortality database found nearly 100,000 deaths from MPM in 83 countries with roughly 5 deaths per million. In addition, men are more frequently affected than women and the average age at death was 70 years old. Asbestos is perfectly safe in its primary state, where it is basically a type of solid rock, but is a significant health hazard when mined or worked in such a way as to produce the carcinogenic nanometer-scale fibrous particles that become an airborne material that is readily absorbed in the lungs. Asbestos fibers can be divided into two main groups: serpentines and amphiboles. Serpentine fibers have only one subtype, which are the chrysotile fibers, that are also called white asbestos due to their light color. These fibers are short and curly and make up around 95% of all commercially used asbestos. Amphibole fibers have many different kinds, including the crocidolite (blue asbestos) and amosite (brown asbestos), tremolite and others. Amphibole fibers are long and straight making them potentially more carcinogenic. Regardless, the International Agency for Research on Cancer (IARC) has classified both fibers equally as Class I carcinogens [10]. There is a known dose-response pattern of asbestos exposure with MM as well as lung cancer, but as stated by the IARC and the WHO, no safe lower threshold has been identified. Asbestos may also cause lung cancer and up to 20,000 asbestos-related lung cancers and 10,000 MM are estimated to occur annually across the population of Western Europe, Scandinavia, North America, Japan and Australia, while registrations are not available in areas that still use asbestos such as Eastern Europe, South America, Africa and the rest of Asia, including China [11,12]. Moreover, MPM has been associated with exposure to erionite, a zeolite mineral with some physical properties similar to asbestos which is widespread in some villages in Cappadocia (Turkey) and some areas of North America [13]. Comparably, a cluster of deaths from pleural mesothelioma has been reported for Biancavilla (Sicily), in Italy. Subsequent studies demonstrated that those MPM cases were related to the patient exposure to fluoro-edenite, a material extracted from quarries which features morphology and composition like that of minerals of the tremolite-actinolite series [14,15]. Thus, the awareness of the potential danger of new man-made and biopersistent fibers with similar carcinogenic properties, exemplified by carbon nanotubes should be strictly monitored to avoid novel epidemic [16]. Though asbestos is certainly the largest and most well-known cause of mesothelioma, roughly 20% of patients do not have any known exposure to asbestos. While it’s possible that these patients were unknowingly exposed, genetic analysis and other studies have led to the suspicion that chemicals such as nitrosamines, nitrosureas, potassium bromate, ferric saccharate, as well as genetic predisposition following chronic exposure to biopersistent minerals and radiation therapy [17] are all culprits [12,13,14,15,16,17,18]. Simian Virus 40 (SV40) infection was previously explored as aetiologic agent but it was not proven [19].

## 3. Pathologic Features

### 3.1. Conventional Histo-Pathology

Such variety of clinical and imaging presentation requires diagnosis to rely on immune-histochemical (IHC) markers: a mesothelioma should be positive for at least two mesothelial markers, and negative for at least two carcinoma-related markers [20]. The most effective combination of IHC markers seem to be: calretinin and cytokeratin 5/6 (or WT1) for the positive markers and CEA (carcinoembryonic antigen) and MOC-31, also known as Epithelial Specific Antigen/Ep-CAM (or B72.3, Ber-EP4, or BG-8) for the negative markers [20,21]. Though this is the current gold standard for MPM diagnosis, it is more easily applied to the epithelioid and biphasic subtypes (which comprise 75–95% of all diagnosis), and not the sarcomatous subtype. This is because the sarcomatous histotype is underrepresented in the literature due to its rarity therefore lacking specific markers [22,23,24]. Thus, an important issue is the heterogeneity of tumors (Figure 1). According to conventional morphology, MPM is divided into three main histological subtypes: epithelioid, sarcomatoid and mixed (biphasic), of which epithelioid is the most common. In the consensus statement for the 2017 diagnostic guidelines for mesothelioma, it was noted that a pleomorphic variant of MPM exists in more than 10% of epithelioid tumors, causing cells to behave similarly to the sarcomatoid and biphasic variants [25]. In ambiguous cases, a rare transitional mesothelioma (TM) pattern may be diagnosed by conventional pathology either as epithelioid, biphasic or sarcomatoid MPM. Morphologic characteristics that favor transitional pattern include sheet-like growth of cohesive, plump, elongated epithelioid cells with well-defined cell borders and a tendency to transition into spindle cells [26]. Detection of homozygous deletion of the *CDKN2A*(*p16*) gene compared to *BAP1* loss through fluorescence in situ hybridization (FISH) on the spindle cell component could be useful to separate ambiguous cases from benign florid stromal reaction and distinguish true sarcomatoid component of biphasic MPM [27]. Very recently, RNA sequencing unsupervised clustering analysis revealed that TM grouped together and were closer to sarcomatoid than to epithelioid MPM [28]. Thus, rather than being separate histological entities, some authors theorize that the mutated cells of MPM progress according to the epithelial-to-mesenchymal transition (EMT). Under this model, epithelioid MPM is epithelial, sarcomatoid MPM is mesenchymal and biphasic MPM is in between the two. Interestingly, long non-coding RNA (lncRNA) fragments have been shown to play diverse roles in EMT and in aggressiveness of MPM and differential signatures which could distinguish between epithelioid and sarcomatoid differentiation have been reported [29]. This theory has been supported by the worse prognosis associated with the sarcomatoid histotype as they are more differentiated from the original epithelium. Part of this switch involves the loss of important markers and regulators of cell function such as E-cadherin and β-catenin. Understanding the classification has diagnostic and prognostic importance, especially with the advent of genomic-based data. For example, Reyniès and colleagues used hierarchical clustering of transcriptomic data to divide MPM (108 frozen tumor samples) into two groups C1 and C2 based on the presence of epithelial and mesenchymal markers [30]. The C1 group corresponded to the histological classification of epithelioid MPM, while the C2 group contained epithelioid, biphasic, sarcomatoid and rarer, undifferentiated types. As expected, the C1 group was associated with a better prognosis than C2. This work demonstrates the importance of taking in mind that certain MPMs with a seemingly epithelioid histotype (theoretical less aggressive behavior) had the underlying genetics of a more aggressive tumor. Epithelial-to-mesenchymal transition (EMT) results in physiological and phenotypic changes which allow epithelial cells to acquire a mesenchymal phenotype. The molecular basis of EMT involves multiple changes in expression, distribution and/or function of transducers, including extracellular matrix and plasma membrane proteins such as periostin, vimentin, integrins, matrix metalloproteinases (MMPs) and cadherins, as well. Transforming Growth Factor β (TGF-β) plays a crucial role in promoting EMT. Indeed it has been reported in vitro that asbestos might induce EMT by downregulating the expression of epithelial markers (E-cadherin, β-catenin, and occluding), and contemporarily, by upregulating mesenchymal markers, such as fibronectin, α-SMA (Alpha-smooth muscle actin), and vimentin [31]. However, the exposure of MPM cells to growth factors such as FGF2 (Fibroblast Growth Factor 2) or EGF (Epidermal Growth Factor) can induce a fibroblastoid morphology, associated to invasive properties, namely scattering, decreased cell adhesion and increased invasiveness. This behavior is mainly related to Mitogen-Activated Protein (MAP)-kinase pathway activation and quite independent of TGF-β or Phosphoinositide-3 (PI3)-kinase signaling [32]. Subsequent microarray analysis demonstrated differential expression of MMP1 (Matrix metalloproteinase-1), ESM1 (Endothelial cell-Specific Molecule 1), ETV4 (ETS Variant Transcription Factor 4), PDL1 (Programmed Death-Ligand 1) and BDKR2B (Bradykinin Receptor B2) in response to both growth factors and in epithelioid vs sarcomatoid MPM. A protein expression analysis on tissue microarray from 352 MPM samples, demonstrated that High expression of membranous EGFR (Epidermal Growth Factor Receptor), integrin β1 and nuclear p27 correlated with epithelioid differentiation whereas high expression of cytoplasmic tumoral and stromal periostin with the sarcomatoid histotype [33]. Notably low expression of periostin in the tumour cell cytoplasm were found to be independent factors for better overall survival. Similarly, high expression of PTEN (Phosphatase and tensin homolog), which is known to be implicated in EMT in cancer [34], acts as positive prognostic factor. EMT is also mediated by hypoxia inducible factor 1 (HIF-1α) through expression of EMT transcription factors such as SNAIL, SLUG, and TWIST1 [35]. In a similar fashion, by modulating cadherin activation, acts mesothelin, which expression is able to promote an EMT-associated phenotype in MPM cells [36]. Moreover, calretinin, a Ca^2+^-binding protein, is implicated in inducing EMT, through the increase of focal adhesion kinase (FAK) expression and/or FAK phosphorylation in MPM cells [37]. Calretinin (CR), through a feedback loop, negatively regulates septin 7, an essential cellular component implicated in the final steps of cell division, as a strong Bt-dependent gene regulatory protein binding to the promoter of *CALB2* [38]. Thus, both CR and septin 7 represent active transducers in MPM genesis and promising actionable targets, as well. Notably, those cells show a higher resistance to cisplatin due to increased Wnt signaling [39].

### 3.2. Genetic Aspects

In light of the lack of mechanistic explanations and sparse treatment options, genetic analysis is providing new hope in the fight against MPM. The genetics of MPM are complex and currently under investigation. This complexity is largely due to the uncommon genetic aberrations and inter-patient variability [5]. Mutations in oncogenes known to be driver of epithelial-derived solid cancers are extremely rare as next generation sequencing (NGS) deep studies have shown [40,41,42] although sometimes with small number of samples (22 MPMs and matched blood samples) [43] and they seem to be not prognostically relevant in the disease [44]. Most alteration found affected the p53/DNA repair and phosphatidylinositol 3-kinase pathways [45], as well as genes involved at transcription level or expression data, such as *SETDB1* [46]. The mutational landscape of MPM is with a signature consistent with production of reactive oxygen species (Figure 1). Notably, asbestos is able to induce chromosome damage and genomic DNA region losses [47,48]. Many reports have confirmed that *TP53* and *RB* tumor suppressor genes are important in maintaining genetic homeostasis in MPM [5]. Although TP53 is vital for the integrity of the genome and thus its mutation can lead to a variety of cancers, *TP53* mutations do not characteristically lead to MPM. Instead, loss-of-heterozygosity (LOH) analysis demonstrated that the two areas of the chromosome most frequently altered in MPM are *CDKN2A–ARF* at 9p21, and *NF2* at 22q12 [5,49,50]. *CDKN2A* encodes for a cell-cycle regulator mutated in more common cancers like melanoma, and neurofibromatosis 2 (NF2) acts as tumor suppressor that is part of the NF2/Merlin complex that makes up the NF2/Hippo pathway [51,52]. Deletions of 3p21 region, enclosing *BAP1* gene are also reported in 33 MPM bioptic samples [53]. LOH of the entire 3p21 region has been reported, whereas many of the deletions described were not contiguous, but rather they alternated along normal DNA segments, as in chromothripsis [54]. Roughly 40% of mesotheliomas have an *NF2* mutation, causing hypophosphorylation of the YAP (Yes-Associated Protein) transcriptional coactivator. YAP is active in its hypophosphorylated state, where it causes transcriptional activation of cell proliferation genes like cyclin D1 (CCDN1) and growth factors like connective tissue growth factor (CTGF). TAZ, encoding for Tafazzin, is a paralog to YAP, two major effectors of Hippo pathways. MPM is one of a few cancers (it has been demonstrated on 12 out of 14 MPM samples) that harbor mutations in Hippo pathway genes [55]. Thus, it would be expected that some patients with mesothelioma would show an activating mutation in *TAZ*, however in vivo evidence of such mutations is lacking. Absence of activating mutation for *TAZ* is also true for *YAP* [56]. BAP1 is a nuclear protein that regulates nuclear material, cell differentiation, gluconeogenesis, transcription, apoptosis. A germ-line mutation in *BAP1* is thought to cause a syndrome that includes mesothelioma, uveal and cutaneous melanoma as well as other neoplasms [57]. When several family members all get mesothelioma, despite only one member of the family working near asbestos, the high incidence can be explained by the transmission of the fibers on the skin and clothes of the one family member to the other members; however, genetic analysis suggests BAP1 mutations may induce augmented susceptibility (Figure 1). Interestingly, *BAP1* mutations seem to prime for epithelial MPM more than any other type, which has important implications for screening and prognosis [58]. Signal transduction pathways and growth factors that stimulate cell survival and proliferation are commonly mutated in cancer cells. In MPM, Extracellular Regulated Kinases (ERK)-dependent phosphorylated antigen, c-MET and the mTOR pathway have all been shown to be activated/enhanced [59]. We and others demonstrated that the activation of mTOR pathway is a prognostic factor for MPM, being phospho-mTOR expression associated to poor response to chemotherapy and shorter overall survival [60]. The HGF-receptor MET has been reported to be activated in MPM due to overexpressed [61], not related to the occurrence of MET genetic lesions: (i) *MET* gene amplification are very infrequent [62] and (ii) mutations in *MET* gene were not found in MPM by recent NGS studies supporting that MET mutations are really rare.

Within respect to gene copy number analysis, an interesting paper by Hylebos et al. analyzed an MPM-cohort (85 cases) for which genomic microarray data available through ‘The Cancer Genome Atlas’ (TCGA) and a validation cohort of 21 cases. Losses on chromosomes 1, 3, 4, 6, 9, 13 and 22 and gains on chromosomes 1, 5, 7 and 17 were found in at least 25% and 15% of MPMs, respectively. Besides the above described M-associated genes, *CDKN2A, NF2* and *BAP1*, other interesting (and not previously described) genes carried a copy number loss (*EP300, SETD2* and *PBRM1*) and four cancer-associated genes showed a high frequency of amplification (*TERT*, *FCGR2B*, *CD79B* and *PRKAR1A*) [63]. In mice combinatorial deletions of Bap1, Nf2, and Cdkn2a result in aggressive mesotheliomas, defined by stem cell-like potential [64]. Previous analysis from the same group revealed gene rearrangements in other unexpected candidate genes. Among them the mitogen-activated protein kinase kinase 6 gene (MAP2K6), which encodes a kinase that phosphorylates p38 in response to stress and inducing apoptosis. Another interesting candidate was dipeptidyl-peptidase 10 gene (DPP10) which impacts on cell cycle regulation by binding specific voltage-gated potassium channels and modulating their function. Finally, amplification of dihydrofolate reductase gene (*DHFR*) and pterin-4-α-carbinolamine dehydratase 2 (*PCBD2*) an enzyme important in folate metabolism, were detected [65].

Whole transcriptome analysis has been used to identify differential gene expression and clustering predictive and prognostic signatures in cancer. Single nucleotide variants were firstly detected on four MPM frozen samples compared to one lung adenocarcinoma and one normal lung sample through pyrosequencing analysis. They occurred in a number of genes, namely x-ray repair complementing defective repair in Chinese gene (*XRCC6*), ARP1 actin-related protein 1 homolog A, centractin α gene (*ACTR1A)*, ubiquinol-cytochrome c reductase core protein 1 gene (*UQCRC1)*, proteasome 26S subunit, non-ATPase 13 gene (*PSMD13*), PDZK1 interaction protein 1 gene (*PDZK1IP1)*, collagen, type V α 2 gene (*COL5A2*), and matrix remodeling associated 5 gene (*MXRA5*) which encode for proteins that were either previously linked to a possible role in tumorigenesis or were found to be overexpressed in different human tumors [66]. Profiles of alternative splicing events have been also generated, such as those involving actin γ 2 smooth muscle enteric gene (*ACTG2)*, cyclin dependent kinase 4 (*CDK4)*, collagen, type III, α 1 gene (*COL3A1)*, and thioredoxin reductase 1 gene (*TXNRD1*) [67]. The most well-studied species of the non-coding transcriptome are microRNAs (miRNAs) are known to modulate gene expression in cancer. With respect to MPM, miR-30b was found to be overexpressed in MPM and locates to 8q24, a frequently accessed region in mesothelioma. Likewise, miR-34 and miR-429 located at 1p36, as well as miR-203 located at 14q32, were not expressed in tumor samples and represented regions frequently affected by DNA copy-number loss [68,69]. Transcriptomic analysis has been also used to assess the differential transcriptional expression of wound-healing-associated genes in MPM during the EMT process [70,71]. Overall, 30 wound-healing-related genes were significantly deregulated, among which potential targets of hsa-miR-143, hsa-miR-223, and the hsa-miR-29 miRNA family members [72]. Out of those genes, *ITGAV* gene expression has been found to display prognostic value, been associated to lower overall survival. A comprehensive, multi-platform, genomic study of 74 MPM samples, as part of The Cancer Genome Atlas (TCGA) showed that poor prognosis subset showed higher aurora kinase A mRNA expression in association with upregulation of PI3K and mTOR signaling pathway [73]. The integrative analysis allowed the identification of prognosis clusters. For instance, poor prognosis signature had a high score for EMT-associated gene expression, which was characterized by high mRNA expression of *VIM*, *PECAM1* and *TGFB1*, and low miR-200 family expression. These tumors also displayed *MSLN* promoter methylation and consequent low mRNA expression of mesothelin, which is a marker of differentiated mesothelial cells, as reported in sarcomatoid MPM and the sarcomatoid components of biphasic MPM [74]. Interestingly, the mRNA expression of *VISTA* (V-domain Ig suppressor of T cell activation), a negative immune checkpoint regulator primarily expressed on hematopoietic cells [75], was strongly inversely correlated with EMT score, being *VISTA* mRNA levels were highest in the epithelioid subtypes. Moreover, an unsupervised analysis of RNA-sequencing data of 284 MPMs identified a continuum of molecular profiles associated to disease prognosis. In particular, immune and vascular pathways emerged as the major sources of molecular variation, and specific profiles were detected: a *hot* bad-prognosis profile, with high lymphocyte infiltration and high expression of immune checkpoints and pro-angiogenic genes; a *cold* bad-prognosis profile, with low lymphocyte infiltration and high expression of pro-angiogenic genes; and a *VEGFR2+/VISTA+* better-prognosis profile, with high expression of immune checkpoint *VISTA* and pro-angiogenic gene *VEGFR2* [76].

It is well known that asbestos induces MPM also involving indirect effects, mainly oxidative stress associated to reactive oxygen species production and DNA-damage. These processes ultimately increase mutation rates and promote malignant transformation [77]. ROS (Reactive Oxygen Species) exposure induces methylation of the gene promoter via a specific recognition site to which DNMT1 (DNA methyltransferase 1) and PARP1 (Poly(ADP-Ribose) Polymerase 1) are recruited, linking DNA damage and DNA methylation. Prolonged ROS exposure induces demethylation by oxidizing the 5-methylcytosine to produce 5-hydroxymethylcytosine, which is catalyzed by ten-eleven translocation methylcytosine dioxygenase (TET) family of enzymes. Hypomethylation of genomic DNA is associated with genomic instability, which in combination with genetic alterations (chromosome deletion), contribute to malignant transformation [78,79,80]. These changes entail DNA oxidation events, post-translational modifications of histones proteins, and DNA methylation. Exposure to asbestos might affect miRNAs expression through epigenetic regulation: a first example regards miR-126. Its expression increases as an adaptive response to asbestos exposure and may proceed to the loss of its expression because of DNA damage accumulation and chromosome deletion, thus leading to carcinogenesis [81]. Interestingly, miR-103 was reported to be significantly down-regulated in the blood cell fraction of 23 patients with MPM, compared to 17 subjects formerly exposed to asbestos, and 25 healthy controls. The differential expression allowed discriminating between MPM patients and asbestos-exposed controls with a sensitivity of 83% and a specificity of 71% [82]. Similarly, the expression of miR-625-3p was reported to be significantly higher in plasma/serum of 30 MPM patients and allowed to discriminate between cases and controls, defined as 14 healthy subjects and 10 subjects with asbestosis [83]. More recently, MPM-specific RNA-based biomarker panels have been detected including DNA damage regulated autophagy modulator 1 (DRAM1) and arylsulfatase A (ARSA), together with their epigenetic regulators: the microRNA (miR-2053) and the lncRNA RP1-86D1.3 [84]. Overall, these circulating signatures should have important features such as low invasiveness and high specificity, which could play a critical role in next future early detection of MPM [85]. These findings give rise to novel attention to availability of compounds that modulate epigenetic modifications, such as histone acetylation or DNA methylation in therapeutic perspective.

### 3.3. Asbestos-Induced Carcinogenesis Mechanisms

The current concept is that tumor grows in a loco-regional pattern, spreading from the parietal to the visceral pleura and invading the surrounding structures that induce the clinical picture of pleural fluid, pain and dyspnea. Distant spreading and metastatization is rarely observed and most patients die from the local growth of the primary pleural mass. MPM has a uniquely moderate mutational landscape which appears to derive from a selective pressure operated by the environment [86]. Mesothelial cells are found in both the parietal and visceral pleura where they can sound the inflammatory alarm in the presence of pro-inflammatory material like asbestos fibers. It’s believed that when fibers are inhaled, they travel through the airways directly to the visceral pleura or arrive there via the lymphatic system [87]. Different fiber types are found in different areas of the lung, where they can disrupt the phagocytosis of the mesothelial cells, causing cytotoxicity, damaging DNA through oxidative stress and thus lead to an inflammatory response. The fibers themselves can cause aberrant separation of chromosomes during mitosis and direct activation of tyrosine kinase receptors, in absence of driver mutations [88,89]. Support that inflammation is a key aspect of MPM pathogenesis is that the lungs contain small milky spots of lymphoid patches (*Kampmeiere’s foci*) on the basal parietal pleura, which is the site where mesothelioma most frequently occurs. Preference for the parietal pleura over the visceral pleura may be rooted in differences in gene expression [5]. Finally, DNA repair/checkpoint genes are important for cancer genesis; these include *BRCA2*, *TOP2A*, *BIRC5* (surviving) and *CHECK1* (checkpoint kinase 1) as demonstrated on tissue microarray comprising 335 MPM patients [90]. Genetics may also point to the reason why some patients are exposed to asbestos and never develop MM, while a small percentage will. Within the pleural space, fibers can cause irritation and repeated cycles of tissue damage. The endpoint of asbestos-induced damage is the generation of an inflammatory microenvironment that may support tumor growth in individuals with predisposition, for instance due to loss of BAP1. Panou et al. demonstrated, by analyzing samples from 198 patients, that a significant proportion of them carry germline mutations in cancer susceptibility genes, as *BAP1*, *CDKN2A*, *TMEM127*, *VHL* and *WT1* [7]. Moreover, it appears that mutations in two genes involved in DNA repair, *XRCC1* and *XRCC3*, along with the GSTM1 (Glutathione S-Transferase Mu 1) antioxidant/detoxifying protein, increase susceptibility in this patient population, as demonstrated on more than 220 MPM samples and matched controls [91,92]. Researching the genetic basis underlying MM is fundamental to risk stratification, diagnosis, prognosis and, especially, the development of novel treatment strategies.

### 3.4. Inflammatory Microenvironment

Inflammation, like that created by asbestos, can prime the cellular terrain, creating a selective pressure that favors cells with an aggressive phenotype (Figure 1). Thus, inflammatory signaling molecules are also frequently enhanced in cancers, and MM is no exception. In mice, xenografts of human mesotheliomas cause inflammation before the development of the tumor [93]. Interleukins 1, 6 and 10, growth factors such as G-CSF, (HGF-Hepathocyte Growth Factor)/scatter factor, and vascular endothelial growth factor (VEGF), and chemokines like CCL2 (C-C motif ligand 2), CCL5, CXCL1 (C-X-C Motif Chemokine Ligand 1) and IFN-γ have all been increased/implicated in MPM pathogenesis. The High Mobility Group Box 1 protein (HMGB1) is a damage-associated molecular pattern (DAMP) protein and a key mediator of inflammation. It is involved in the early stages of mesothelial cell transformation upon exposure to asbestos and erionite by establishing an autocrine circuit influencing cell proliferation and survival [94]. Bianchi and colleagues demonstrated that under severe cellular stress, HMGB1 is relocated from the nucleus to the cytoplasm and then to secretory lysosomes or directly to the extracellular space. Then, the extracellular space triggers inflammation and adaptive immunological responses by switching among multiple oxidation states. Moreover, HMGB1 supports tissue repair and, by coordinating the switch of macrophages to a tissue-healing phenotype, activation and proliferation of stem cells and neoangiogenesis. Concomitantly, it enhances the immunogenicity of mutated proteins in the tumor (neoantigens) thus promoting anti-tumor responses [95]. Overall, inhibiting HMGB1 by a HMGB1 monoclonal antibodies (mAb), by the recombinant HMGB1 antagonist BoxA and by a mAb against the HMGB1 main receptor RAGE (Receptor for Advanced Glycation Endproducts) impair MPM progression in vitro and in animal models [96]. HMGB1 functions as a ‘master switch’ by which the chronic inflammation that drives mesothelioma growth is initiated and maintained. Overall HMGB1 plays a crucial role in MPM onset and progression according to the following mechanisms: (i) asbestos-induced effector since its secretion by mesothelial or immune cells is highly responsive to asbestos fiber stimulation; (ii) inflammatory and epithelial-to-mesenchymal transition mediator. For instance, it induces tumor necrosis factor-α secretion by macrophages thus triggering chronic peritumoral inflammation [97]. Moreover, HMGB1 can increase the expression of cadherins thus promoting cellular mesenchymal differentiation associated to malignant phenotype [98]. In this perspective the serum level of HMGB1 is considered to be a predictive biomarker for monitoring occupational workers and subjects at higher risk to develop MPM, although preliminary reports have been conducted in limited population [99,100]. Notably, it has been reported that therapeutic levels of aspirin and its metabolite salicylic acid can suppress growth, migration, invasion, wound healing, and anchorage-independent colony formation of HMGB1-secreting human mesothelioma cells [101,102]. Moreover, a number of inflammatory cells can be found in MPM-surrounding stroma. The vast majority (30%) is represented by macrophages expressing markers (like CD206) associated with tissue-healing phenotype M2. In contrast M1 macrophages (classically activated macrophages) shows pro-inflammatory, tissue destructive and anti-tumor activity. Tumor associated macrophages (TAMs) derive from circulating monocytic precursors. In MPM, monocytes are recruited by several chemokines and interleukines, such as IL-4, IL-13 and IL-10 produced by tumor infiltrating lymphocytes (TILs). The latter promote differentiation of macrophages towards an M2 phenotype [103]. A high ratio of intratumoral M2 macrophages is a negative prognostic factor in epithelioid MPM [104]. Complex T cell infiltrates are generally found. Myeloid-derived suppressor cells (MDSCs) CD33 and CD11b positive induce Tregs and produce nitric oxide and arginase, leading to loss of function of CD4+ and CD8+ T cells [105]. This immunosuppressive milieu ultimately promotes immune escape, tumor growth, invasion and angiogenesis [106]. Higher levels of TILs have been associated with better survival in MPM. Notably, numbers of CD45+ leukocytes were increased in non-epithelioid mesothelioma compared to epithelioid ones and seem to be associated with worse response to chemotherapy [107,108]. Higher fraction of FOXP3+ (Forkhead boX P3)/CD4+ Tregs have been reported in MPM, both chemotherapy-pretreated and untreated and is associated to worse prognosis [109]. In general, PD-L1-positive cells are heterogeneously present in MPM, being PD-L1 expression higher in non-epithelioid mesothelioma compared to epithelioid mesothelioma [110]. Dissecting the properties of these inflammatory cells within tumors will provide greater insights into the immunologic mechanisms of response and resistance to immunotherapy in this disease. Awad and colleagues showed, by applying flow cytometry to characterize 43 resected MPM specimens, distinct immunologic phenotypes in PD-L1–positive tumors as compared with PD-L1–negative ones, and in sarcomatoid/biphasic tumors vs epithelioid ones. Frequencies of T cells in the 38-patient cohort were highly variable, but showed a similar differentiation status and cellular composition, including a relatively high percentage of CD4+ T cells that expressed FOXP3 (~20%). In detail they found that PD-L1–positive and sarcomatoid/biphasic tumors have a significantly greater proportion of infiltrating T cells than PD-L1-negative and epithelioid tumors, respectively [111]. PD-L1-positive tumors also show significant increases in T-cell proliferation and activation, along with significant increases in Tregs and expression of T-cell-inhibitory markers, such as TIM-3 (T cell immunoglobulin domain and mucin domain -3) [112]. The work by Klampatsa et al. extended the findings by analyzing fresh tumor and blood samples of 22 MPM cases and demonstrated high levels of the inhibitory receptor TIGIT (T cell immunoreceptor with Ig and ITIM domains) (~60%), CD39 (~20%), and CTLA4 (Cytotoxic T-Lymphocyte Antigen 4) (~25%) [113]. Overall, they found that MPM TILs were consistently hypofunctional, mainly associated with higher numbers of CD4 regulatory Tregs and with expression of TIGIT. Although the TILs showed uniformly high levels of cytokine production. The considerable immunophenotypic variability is coherent to the variable responses obtained in MPM by PD-L1 inhibitors [114], although other factors are involved: (i) the abundance of infiltrating lymphocytes; (ii) co-expression of multiple inhibitory receptors on T cells; (iii) the influence of MDSCs and tumor-associated macrophages [115].

### 3.5. Serology

Non-invasive measures are coming to the surface and involve the analysis of various serum proteins using a 13-protein classifier as well as individually looking for soluble mesothelin-related peptides (SMRP), mesothelin, fibulin-3 and micro-RNAs in the plasma. The sensitivity and specificity of SMRP/Mesothelin has been debated in the literature but most studies seem to suggest that sensitivity and specificity are great enough to make them acceptable markers of tumor burden; they also correlate with disease severity as they’re most elevated in late disease states [5,116,117,118]. Using the Slow Off-rate Modified Aptamers (SOMA)-scan proteomic assay, a highly sensitive candidate 13-biomarker panel was discovered and validated (on 117 MPM samples and 142 asbestos-exposed control individuals) for the detection of MPM in the asbestos-exposed population with an accuracy of 92% and detection of 88% of Stage I and II disease [119]. The 13-protein classifier uses short segments of DNA-like molecules that can bind to molecular targets and capture various inflammatory and proliferative proteins found in the serum when a patient is positive for mesothelioma. This method has promising results and potential usefulness in surveillance and diagnosis of MPM in those subjects at highest risk for the disease, but it has still not been validated for diagnosis. Fibulin-3 is an extremely sensitive and specific marker found in the plasma and in pleural effusion fluid that can identify patients with mesothelioma from those without it [120]. Osteopontin (OPN) is a glycoprotein known to be overexpressed in several human cancers. Interestingly, high circulating levels of OPN have been detected in samples from 56 MPM patients and serum OPN level may act as useful diagnostic marker for MPM patients [121]. More recently, the expression of brain-derived neurotrophic factor (BDNF), a neurotrophin, has been demonstrated in MPM. In detail high BDNF expression, at the mRNA level have been reported in tumors and at the protein level in pleural effusions (PE), thus becoming as a novel specific hallmark of MPM samples. Notably, high *BDNF* gene expression and PE concentration were predictive of shorter MPM patient survival in a cohort of 79 MPM tumor samples and 26 normal pleura. Moreover, BDNF activation is implicated in the PE-induced angiogenesis: this observation has potentially strong clinical implication and supports rationale of targeting angiogenesis in MPM [122]. Circulating cell-free RNA (MirRNA) fragments might serve as biomarkers in several diseases. Within respect to MPM, the MiR-625-3p has been found in the serum which has high stability and thus high diagnostic potential in a cohort of 15 MPM samples matched with 14 normal cases, weather the validation cohort was defined by 30 MPM and 10 asbestosis cases, respectively [83]. Its use as a specific and sensitive diagnostic marker is still under investigation. These markers, along with genomic data, as mentioned before, are shedding light on the possibility of new classifications and diagnostic methods.

## 4. Diagnostic and Therapeutic Approaches

### 4.1. Current Diagnostic and Staging Systems

Diagnosis of MPM is difficult as the clinical presentations and imaging are variable and non-specific. Imaging usually begins with chest X-ray (CXR) and then moves on to computed tomography (CT). Rarely, magnetic resonance (MR) is used for staging. All imaging features are detailed and summarized in Table 1. CT characteristics that are suggestive of MPM are similar to those in CXR but are greater in number and detail. TNM (Tumor Nodes Metastases) staging, and thus prognosis, is also possible through CT images evaluation. Novel approach such as quantitative textural and shape analysis (radiomics signature) may help distinguish malignant from benign lesions [123]. Although CT is the mainstay technique, MR, PET-CT and ultrasound can all provide important additional information. PET and PET-CT can help distinguish benign and malignant diseases, based on the standardized uptake value (SUV), and also aid in decision-making regarding staging and potential treatment options. Ultrasound (US) is especially important to analyze pleural effusion. In addition, US is vital in guiding the needle placement in thoracentesis and drain placement (the main treatment of advanced MPM cases under palliative care), as well as needle biopsies. US has the disadvantage of being extremely user-dependent, however, expert sonographers can still obtain valuable information for diagnosis and treatment. The undoubted advantages of US are that it is quick, painless, and inexpensive.

Thoracoscopy is performed in the diagnostic phase as standard procedure that allows a good visualization of the endothoracic anatomy with direct evaluation of the locoregional tumor extension and to practice, where necessary, an effective chemical pleurodesis. Medical thoracoscopy is a safe procedure to obtain histological diagnosis and must include at least five biopsies performed on the pathological pleura by sampling the lesions and possibly also the apparently normal pleura in a representative manner. Biopsies on the parietal pleura must be deep enough to allow evaluation of the invasion of fat and muscle of the chest wall [132,133]. In selected cases and, mandatory in the case of complex pleural spaces (e.g., low and loculated effusion) the approach of choice is the video assisted thoracoscopy (VATS) that allows practicing other pleurotomies for use straight optics without an operating channel and other essential tools for a more complex procedure. The diagnostic sensitivity of thoracoscopy is extremely high, reaching percentages of 98% [134]. Notably, the thoracoscopic approach may also be required for pleurodesis in the advanced stages, as obliteration of pleural space allows control of relapsing or massive pleural effusion [135,136,137].

### 4.2. Current Treatments and Targeted Agents

Unfortunately, few treatment options exist for MPM, though not for a lack of volition nor attempts. The 2018 guidelines of the National Comprehensive Cancer Network determine that the first-line systemic therapy for MPM is pemetrexed paired with cisplatin and possibly bevacizumab [138,139]. Bevacizumab is a humanized monoclonal antibody that inhibits vascular endothelial growth factor (VEGF), a key growth factor in MPM pathology. Addition of bevacizumab to pemetrexed plus cisplatin has been demonstrated to significantly improve overall survival (OS) in a cohort of 448 MPM patients [140]. Second-line therapy includes vinorelbine, gemcitabine and various biological therapies [141]. Surgical therapy, in terms of pleurectomy/decortication, is only for patients who can handle the surgical risks and who have a low tumor stage. This cytoreduction may involve the partial removal of tumor mass and pleura if it can significantly improve the quality of life of the patient without creating excessive morbidity. Particular attention should be paid to lymph node involvement; they should be sampled during the procedure and their positivity drastically reduces survival [62,134,142]. Radiation therapy (RT) can have a role in MPM treatment as well, as it can be in conjunction with chemotherapy and can provide local tumor control if the patient has a good performance status. Toxicity, however, may be significant and thus RT alone has little benefit unless it aims to relieve specific symptoms such as chest pain or bronchial/esophageal obstruction. In cases where systemic, surgical and radiological treatment are no long indicated or not successful, supportive care is the mainstay of treatment.

### 4.3. Treatment Failure and Further Investigation

Modern targeted therapies that have shown benefit in other human tumors—e.g., lung cancers—have so far failed in MPM. Overall, this has led to MM being listed an orphan disease by the European Union (EU). The need for new therapies is therefore urgent. New treatment options are necessary to combat MM as most traditional chemotherapies have failed to significantly improve survival, and expensive monoclonal antibodies only increase survival a few months at best. The list of modern chemotherapies that have been tried and deemed unsuccessful includes tyrosine kinase inhibitors like erlotinib and gefitinib and the mTOR inhibitor, everolimus [143,144,145]. In addition, immune checkpoint inhibitors like the anti-PD1 antibody pembrolizumab and the anti-PD-L1 antibody durvalumab have been unsuccessful [146].However, very recently it has been reported that anti-PD-1 nivolumab monotherapy or nivolumab plus anti-CTLA-4 ipilimumab combination therapy both showed promising activity—without unexpected toxicity—in 125 MPM patients affected by relapsing disease after first-line or second-line pemetrexed and platinum-based treatments [147]. In 2017, the phase 2b, multicenter, randomized, double-blind, controlled trial investigated tremelimumab, a CTLA-4 inhibitor, as second or third-line therapy in 569 patients with relapsed MPM and found no statistically significant impact on overall survival [148]. Overall MPM is characterized by a strong immunosuppressive component and this issue is implicated in the relatively low response rates to checkpoint inhibitors. On this basis, novel immunotherapeutic strategies are under investigation. In MPM patients, dendritic cells (DCs) have been shown to be reduced in numbers and in antigen-processing function compared to healthy controls and negatively affected survival outcomes [149]. On this basis, DC vaccination represents a promising therapeutic strategy. In nine cases, DC immunotherapy using allogeneic MPM tumor lysate have been associated to enhanced frequencies of B cells and T cells in blood [150]. The DENdritic cell Immunotherapy for Mesothelioma (DENIM) trial has been designed to evaluate the efficacy of autologous DCs loaded with allogenic tumor lysate (MesoPher) in MPM patients after first line treatment with chemotherapy MPM patients have been randomized to receive either DC therapy plus best supportive care (BSC) or BSC alone, with overall survival as primary end point [151]. Although the trial is still ongoing, preliminary results demonstrate that DC therapy seem to be safe and effective as a maintenance treatment and promising novel treatment option. Another approach that has been investigated in MPM regards the administration of tumor antigen-targeted T cells with transduction of a chimeric antigen receptor (CAR). CARs are synthetic receptors that enhance T-cell antitumor effector function. Growing experimental and clinical evidence sustains the therapeutic role of CARs not only in the treatment of hematologic malignancies but also in solid tumor, including MPM [152]. Interestingly, overexpression of the MET receptor can be effectively targeted using ligand-directed CAR T-cells. Indeed, MET re-targeted CAR T-cells exert in vivo anti-tumor activity against an established mesothelioma xenograft [153]. Quite surprisingly, local delivery of targeted CAR-T seems to be highly promising. This hypothesis is allowed by the experimental observation that mesothelin-targeted CARs induce antitumor activity when subcutaneously injected in mod animal models of MPM and that intra-pleural T cell administration is vastly superior to systemic infusion in orthotopic MPM models [154]. Another promising approach regards manipulation of induced pluripotent stem cells (iPSC) which express tumor-associated antigens vaccine promotes an antigen-specific anti-tumor T cell response. iPSC vaccines prevent tumor growth in syngeneic models of solid cancers including MPM by promoting an antigen-specific anti-tumor T cell response [155].

Recent insights regarding epigenetic alterations in MPM provide the preclinical rationale for design of targeting epigenome in MPM [156]. Among the most relevant examples, there is belinostat. It is a histone deacetylase (HDAC) inhibitor that targets the epigenetic pathway also showed no effect in 13 cases affected by progressing disease [157]. However, the arsenal of potential therapies is vast, and some promise is on the horizon. MPM cells are less capable of synthesizing arginine, thus depleting available arginine with pegylated-arginine deiminase (ADI-PEG20) prolonged progression free survival (PFS) in one study performed after a screening of 201 patients which allowed the identification of 68 with advanced ASS1-deficient malignant pleural mesothelioma [158]. As previously mentioned, *BAP1* is a tumor suppressor and mutations in *BAP1* are important in MPM pathogenesis, especially in MPM associated with familial clusters. When BAP1 is lost, EZH2, a protein that participates in histone methylation, increases, which has a role in proliferation of malignant cells. In animal studies on MPM with *BAP1* mutations, inhibiting EZH2 halted MPM progression, thus posing a potential therapeutic target [38,159]. Continuing with treatments in the epigenetic pathway, the DNA methyl transferase (DNMT) inhibitor dihydro-5-azacytidine had a small clinical effect in MPM in 29 evaluated cases [160] in absence of correlation to the presence of p16INK4a methylation [161]. Monoclonal antibodies against mesothelioma markers, mesothelin and CD26 are also current research topics that show some promises [162,163]. Statins, a common and inexpensive treatment for dyslipidemia and cardiovascular disease prevention, have been thought to have a potential inhibitory effect on MPM cells, especially when combined with doxorubicin, based on laboratory studies. Regarding the NF2/Merlin pathways, it is thought that statin’s inhibitory effect on the mevalonate pathway also causes them to inhibit the YAP/TAZ transcription proteins [164]. Another common drug that acts on metabolic pathways is metformin. Retrospective studies showed it does not change patient survival alone; however, when paired with nutlin 3a, a protein that prevents TP53 degradation, it has been shown to inhibit MPM proliferation [165]. Apart from statins, other potential therapies targeting the NF2/Merlin pathway include MLN4924, which is an enzyme that stops the activation of YAP1 in MPM cells with NF2 mutations. MLN4924 plus an mTOR/PI3K inhibitor showed promising effects in both in vitro and in vivo studies [166]. C19 is another molecule that can act on this pathway by degrading TAZ. This small molecular inhibitor, as well as statins and the other above-mentioned inhibitors, are still being researched but could provide promising results [167].

Tumor heterogeneity can importantly affect drug penetration and distribution on one hand [168] whereas on the other, activation of resistance mechanisms can be associated to the inflammatory tumor microenvironment. On this basis, a novel therapeutic approach is directed to ameliorate drug delivery (and efficacy) towards tumor mass. Specific imaging methods have been used to study paclitaxel distribution in several cancer cells, among which MPM cell line [169]. Similarly, the 3D cultures models of MPM give important information about the drug concentration [170]. Moreover, since pleural space provides a unique opportunity for local treatment, we reported interesting data about drug-loaded nanocarriers targeted towards CD146, specifically expressed by primary cell lines obtained from MPM effusions [171]. In detail, gold nanoparticles vehicling conventional chemo (pemetrexed [125]), or biological agents [172] were more active than drugs alone in inhibiting in vitro malignant phenotype. Interestingly, the adhesion molecule CD146 is expressed in a variety of cancers and in MPM but not in reactive mesothelial cells [173]. The fibroinflammatory stroma typical of MPM, can contribute to chemoresistance by stimulating cancer cells growth, invasion, and angiogenesis, and inducing an immunosuppressive phenotype, as discussed above. Thus, the immune suppressive microenvironment in mesothelioma is likely to be involved in the poor response to novel immunotherapies, if compared to other solid cancers [174]. Moreover, strong preclinical evidences support a role of hypoxia and MPM cancer stem cells (CSCs) in determining tumor resistance to therapies. Indeed, it has been reported that MPM contains hypoxic regions [175,176] and the hypoxic microenvironment is well known to activate many signaling pathways involved in tumor initiation, progression and maintenance as well as chemo-radio-resistance [177,178]. Hypoxia also modulates gene and microRNA (miRNA) expression, which has been also been associated to stemness [179] and to resistance to therapies [180,181].

## 5. Conclusions and Future Prospects

Although a relevant number of genomic alterations are known to drive epithelial carcinogenesis, very few data have been reported regarding MPM onset. Therefore, at the present, no actionable targets can be exploited to effectively treat MPM. Growing evidence suggests that asbestos-induced inflammation might cause the malignant transformation of mesothelial cells. The unique tumor microenvironment is involved in inducing resistance to therapies that usually characterizes the clinical setting. Promising diagnostic and therapeutic interventions will derive, in the next future, from a deeper understanding of cellular and molecular mechanisms regulating the crosstalk between tumor microenvironment and neoplastic cells. Manipulations of the inflammatory tumor stroma will render MPM susceptible to therapy with checkpoint inhibitors that have shown relevant results in other solid and aggressive tumors as lung cancer.

## Figures and Tables

**Figure 1 cancers-12-01186-f001:**
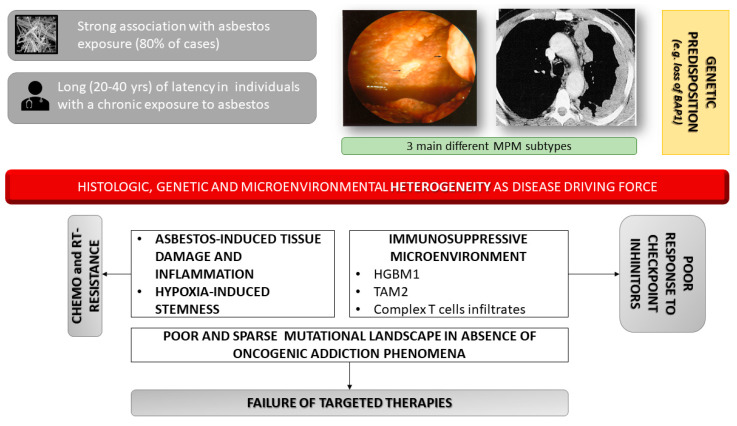
Genetic and inflammatory-immune circuits associated to MPM (Malignant Pleural Mesothelioma) onset and progression. Malignant pleural mesothelioma is mainly associated to asbestos exposure and can develop after long latency with no known dose threshold. Notably, it is likely that some subjects are more genetically susceptible, but absence of dose threshold is not only due to this genetic susceptibility. No driver mutations are known to affect oncogenic drivers. Moreover, the unique tumor microenvironment is involved in inducing resistance to therapies that is usually characterizes clinical setting.

**Table 1 cancers-12-01186-t001:** Imaging features of MPM (Malignant Pleural Mesothelioma). Detection of pathologic findings (in percentages) in the pleural layers and other thoracic districts through chest X rays (CXR) and CT (computed tomography) scan. Percentages of detection of pathologic findings have been obtained from literature revision [124,125,126,127,128,129,130,131].

Imaging Features		CXR (%)	CT Scan (%)
**Pleura**			
	Unilateral effusion	30–80	90–95
	Bilateral effusion	30–40	mar-20
	Mass and thickening	25	38: mass 22: smooth 20: irregular 7: nodules
	Thickened and calcified on the parietal side-plaques	20	20
	No alterations		
**Concomitant Findings**			
	Reduced volume of the involved lung	**P**	**P**
	Mediastinal lymphadenopathy	**P**	**P**
	Destruction of a rib due to local invasion	**P**	**P**
	Shifting of the mediastinum	**P**	**P**

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
