# Peer review of "Malignant Pleural Mesothelioma: Genetic and Microenviromental Heterogeneity as an Unexpected Reading Frame and Therapeutic Challenge"

_cancers, 2020, doi:10.3390/cancers12051186_

Round 1
Reviewer 1 Report
Despite this review is quite comprehensive, I have some concerns about it.
First, even if most of the statements are correct, some of them need to be corrected/clarified.
Second, the authors need to be very cautious with the citations as I notice several references have not been cited correctly. In my major comments, I list several of them, but authors need to review all their references.
Third, for some parts of the manuscript such as “In vivo biological responses” or “Treatment failure and further investigation”, the authors rely on only one or two reviews. The author should go further than the previous reviews.
Major comments
- For the sentence in the introduction, “ Growing evidence […]”, the reference 8 is not relevant. The authors should cite recent articles suggesting that, beside BAP1, other germinal mutations are present in MPM.
- In the “epidemiology “part, authors should mention that asbestos is not ban in some countries as the first sentence is ambiguous and let think that it is worldwide.
- In the “epidemiology” part, “ In addition, men are more frequent […]”, the reference 8 is again not relevant.
- In the asbestos part, “ Regardless, the International Agency for Research on Cancer […]”, the reference 6 could be also replaced by a more adequate reference.
- In the “asbestos” part, fluoro-edenite should be also mentioned
- In the “asbestos” part, reference 22 is cited before reference 16 and is not relevant.
- In the “Other possible etiological agents” part, it seems to me that evidences support radiation involvement in MPM.
- In table 1, the authors should indicate how the percentages of detection of pathologic findings were calculated or what source they come from.
- In the legend of figure 1, the authors stated : “Malignant pleural mesothelioma is mainly associated to asbestos exposure and can develop after long latency with no known dose threshold in genetically susceptible subjects.” It is likely that some subjects are more genetically susceptible, but absence of dose threshold is not only due to this genetic susceptibility. So I think this sentence could be misinterpreted.
- In the “Conventional histo-pathology” part, “This is because the sarcomatous […]”, the reference 31 could be also replaced by a more adequate reference
- In the “Conventional histo-pathology” part, “Specifically, the desmoplastic variant […]”, the percentage of desmoplastic MPM is largely overestimated as desmoplastic is now considered as a subtype of sarcomatoid MPM.
- In the “Conventional histo-pathology” part, “In the consensus 187 statement for the 2017 diagnostic guidelines for mesothelioma, […]”, the reference 23 is wrong and is 33.
- In the “Genetic asset” part, “Mutations in oncogenes known to be driver of […]”, the authors should mention that mutations in oncogene are extremely rare as NGS large studies have shown. This is in line with the statement of the authors in the legend “No driver mutations are known to affect oncogenic drivers.”.
- In the “Genetic asset” part, “The mutational landscape of MPM is sparse - coherently with the absence of a strong direct mutational effect of asbestos - with a signature consistent with production of reactive oxygen species (Fig.1). […]”, I am not completely agree the statement on asbestos. I think the authors should mention the ability of asbestos to induce chromosome damage and genomic DNA region losses. Several articles and reviews pointed out the genotoxicity by itself of asbestos and that need to be mentioned by the authors.
- In the “Genetic asset” part, “Instead, loss-of-heterozygosity analysis […]”, the authors should also mention 3p21 region (enclosing BAP1 gene), for which more recent data than reference 5 showed also LOH and where chromothripsis was highlighted.
- In the “Genetic asset” part, ‘TAZ is a paralog …]”, the reference 28 is not accurate. Actually, absence of activating mutation for Taz is also true for Yap.
- In the “Genetic asset” part, “A germ-line mutation in BAP1 […]” and “Interestingly, BAP1 mutations […]”, there are some specific recent articles and reviews on BAP1 more relevant than reference 5.
- In the “Genetic asset” part, “The HGF-receptor MET has been reported […]”, to my knowledge, amplification and mutation of MET are very unfrequent in MPM.
- In the “In vivo biological responses” part, which looks like more as a paragraph on asbestos carcinogenesis mechanism, authors only cited reference 5 of 2015. For example, they should cite the correct reference for XRCC1 and XRCC3. Furthermore, as I mention previously, other germinal mutations are present in MPM and the recent articles should be mention. Overall, the authors should go further than the review 5.
- In the “Inflammatory microenvironment” part, references 14/16 are not relevant.
- In the “serology” part, a reference should be indicated for the 13-protein classifier and explanations on this classifier should be clarify.
- In the “serology” part, a recent article suggest that BDNF could be an interesting biomarker and should be mentioned.
- In the “Treatment failure and further investigation” part, several references are also inadequately cited such as references 37, 38, 75.
- In the “Treatment failure and further investigation” part, the authors stated “In addition, immune checkpoint inhibitors like the anti-PD1 antibody pembrolizumab and the anti-PD-L1 antibody durvalumab have been unsuccessful”. They should take into account the MAPS2 clinical trials published in Lancet Oncol. in 2019.
- In the “Treatment failure and further investigation”, the authors relied mainly on 1 or 2 reviews. Original articles should be mentioned and they should go further than the previous reviews.
Minor comments
- Define three keywords.
- Line 245, Reference 40 in duplicate.
- Line 277, a bracket is missing.
- Line 283 YAZ instead of YAP.
- Gene names need to be italicized.
- “Treatment failure and further investigation” part is is numbered 6.1 and there is no 6.2.
- Reference 75 was cited before 74.
Author Response
Reply to Reviewer 1
Comments and Suggestions for Authors
Despite this review is quite comprehensive, I have some concerns about it. First, even if most of the statements are correct, some of them need to be corrected/clarified. Second, the authors need to be very cautious with the citations as I notice several references have not been cited correctly. In my major comments, I list several of them, but authors need to review all their references. Third, for some parts of the manuscript such as “In vivo biological responses” or “Treatment failure and further investigation”, the authors rely on only one or two reviews. The author should go further than the previous reviews.
We really thank the Reviewer for this fruitful suggestion which really improve the quality of the manuscript. We have deeply revised and updated the reference section, accordingly. Below the point-by-point answers (A).
Major comments
- For the sentence in the introduction, “ Growing evidence […]”, the reference 8 is not relevant. The authors should cite recent articles suggesting that, beside BAP1, other germinal mutations are present in MPM.
A1. We thank the reviewer for this suggestion, and we added two more references (Panouv V et al J Clin Oncol. 2018; 36(28):2863-2871. doi: 10.1200/JCO.2018.78.5204 and Attanoos RL et al Arch Pathol Lab Med. 2018;142(6):753-760. doi: 10.5858/arpa.2017-0365-RA).
- In the “epidemiology “part, authors should mention that asbestos is not ban in some countries as the first sentence is ambiguous and let think that it is worldwide.
A2. We fully agree with this comment and the text has been modified as follows: “…Though mesothelioma is rare and the production of materials with asbestos has been illegal for more than 20 years in many counties - although it is not ban in some others - the incidence of MPM is still rising…”
- In the “epidemiology” part, “In addition, men are more frequent […]”, the reference 8 is again not relevant.
A3. We removed the reference as indicated
- In the asbestos part, “ Regardless, the International Agency for Research on Cancer […]”, the reference 6 could be also replaced by a more adequate reference.
A4. We agree with this suggestion and we introduced the following reference: IARC Working Group on the Evaluation of Carcinogenic Risks to Humans. Arsenic, metals, fibres, and dusts. IARC Monogr Eval Carcinog Risks Hum. 2012;100(Pt C):11-465.
- In the “asbestos” part, fluoro-edenite should be also mentioned
A5. We really thank the Reviewer for this suggestion. The text has been implemented as follows: “Comparably, a cluster of deaths from pleural mesothelioma has been reported for Biancavilla (Sicily), in Italy. Subsequent studies demonstrated that those MPM cases were related to the patient exposure to fluoro-edenite, a material extracted from quarries which features morphology and composition like that of minerals of the tremolite-actinolite series (Paoletti L et al. Arch Environ Health. 2000;55(6):392-8. doi: 10.1080/00039890009604036 and Comba P et al. Arch Environ Health. 2003;58(4):229-32. doi: 10.3200/AEOH.58.4.229-232)”.
- In the “asbestos” part, reference 22 is cited before reference 16 and is not relevant.
A6. The reference has been removed and the text modified, also coherently to Reviewer 2 suggestion.
- In the “Other possible etiological agents” part, it seems to me that evidences support radiation involvement in MPM.
A7. We thank the Reviewer for this comment and (also based on Reviewer 2 comment) the text has been shortened and implemented as follows: “…While it’s possible that these patients were unknowingly exposed, genetic analysis and other studies have led to the suspicion that chemicals such as nitrosamines, nitrosureas, potassium bromate, ferric saccharate, as well as genetic predisposition following chronic exposure to biopersistent minerals and radiation therapy (Farioli A et al. Cancer Med. 2016 May;5(5):950-9. doi: 10.1002/cam4.656.] are all culprits...”
- In table 1, the authors should indicate how the percentages of detection of pathologic findings were calculated or what source they come from.
A8. We thank the Reviewer for this comment. We obtained the percentages of pathologic imaging findings by analysing and matching literature data. Most percentages are reported in different papers. Thus, we have modified accordingly the legend and introduced all the refences, as follows: “… Percentages of detection of pathologic findings have been obtained from literature revision (Fortin M et al.Respiration. 2020;99(1):28-34. doi: 10.1159/000503239, Escalon JG et al. J Comput Assist Tomogr. 2018 ;42(4):601 606. doi: 10.1097/RCT.0000000000000727,Patz EF et al. Am J Roentgenol 1992 159:5, 961-966 doi: 10.2214/ajr.159.5.1414807, Kim YK et al. Korean J Radiol. 2016 ;17(4):545-53. doi: 10.3348/kjr.2016.17.4.545, Nickell LT Jr et al. Radiographics. 2014;34(6):1692-706. doi: 10.1148/rg.346130089, Cardinale L. et al. Acta Biomed. 2017;88(2):134–142. doi:10.23750/abm.v88i2.5558, Wang Z et al., RadioGraphics 2004, 24:1, 105-119 doi:10.1148/rg.241035058, Tamer Dogan O et al. Iran J Radiol. 2012 ;9(4):209-11. doi: 10.5812/iranjradiol.8764)…”
- In the legend of figure 1, the authors stated: “Malignant pleural mesothelioma is mainly associated to asbestos exposure and can develop after long latency with no known dose threshold in genetically susceptible subjects.” It is likely that some subjects are more genetically susceptible, but absence of dose threshold is not only due to this genetic susceptibility. So I think this sentence could be misinterpreted.
A9. We agree with this comment. The text has been revised as follows: “Malignant pleural mesothelioma is mainly associated to asbestos exposure and can develop after long latency with no known dose threshold. Notably, it is likely that some subjects are more genetically susceptible, but absence of dose threshold is not only due to this genetic susceptibility…”
- In the “Conventional histo-pathology” part, “This is because the sarcomatous […]”, the reference 31 could be also replaced by a more adequate reference
A10. We agree with this comment and the reference has been removed and replaced by: Panou V et al. Cancer Treat Rev. 2015;41(6):486-95. doi: 10.1016/j.ctrv.2015.05.001 and Ascoli V et al. Pathol Res Pract. 2016 ;212(10):886-892. doi: 10.1016/j.prp.2016.07.010).
- In the “Conventional histo-pathology” part, “Specifically, the desmoplastic variant […]”, the percentage of desmoplastic MPM is largely overestimated as desmoplastic is now considered as a subtype of sarcomatoid MPM.
A11. We thank the Reviewer for this comment. The text has been modified as follows: “ …According to conventional morphology, MPM is divided into three main histological subtypes: epithelioid, sarcomatoid (encompassing the desmoplastic subtype which percentage is largely overestimated) and mixed (biphasic), of which epithelioid is the most common...”
- In the “Conventional histo-pathology” part, “In the consensus 187 statement for the 2017 diagnostic guidelines for mesothelioma, […]”, the reference 23 is wrong and is 33.
A12. We are sorry for the mistake and correct the reference, as indicated.
- In the “Genetic asset” part, “Mutations in oncogenes known to be driver of […]”, the authors should mention that mutations in oncogene are extremely rare as NGS large studies have shown. This is in line with the statement of the authors in the legend “No driver mutations are known to affect oncogenic drivers.”.
A13. We agree to this suggestion and the text has been implemented as follows also accordingly to Reviewer 3 comments: “…Mutations in oncogenes known to be driver of epithelial-derived solid cancers are extremely rare as next generation sequencing (NGS) deep studies have shown (Carbone M et al. J Thorac Oncol. 2015;10(3):409-11. doi: 10.1097/JTO.0000000000000466) although sometimes with small number of samples Guo G et al. Cancer Res. 2015; 75(2):264-9. doi: 10.1158/0008-5472.CAN-14-1008) ...”
- In the “Genetic asset” part, “The mutational landscape of MPM is sparse - coherently with the absence of a strong direct mutational effect of asbestos - with a signature consistent with production of reactive oxygen species (Fig.1). […]”, I am not completely agree the statement on asbestos. I think the authors should mention the ability of asbestos to induce chromosome damage and genomic DNA region losses. Several articles and reviews pointed out the genotoxicity by itself of asbestos and that need to be mentioned by the authors.
A14. We thank the Reviewer for this fruitful comment. The text has been implemented as follows: “…Notably, asbestos is able to induce chromosome damage and genomic DNA region losses (Solbes E et al. J Investig Med. 2018;66(4):721-727. doi: 10.1136/jim-2017-000628 and Xu A et al. Environ Health Perspect. 2007;115(1):87-92. doi: 10.1289/ehp.9425) ...”
- In the “Genetic asset” part, “Instead, loss-of-heterozygosity analysis […]”, the authors should also mention 3p21 region (enclosing BAP1 gene), for which more recent data than reference 5 showed also LOH and where chromothripsis was highlighted.
A15. We thank the Reviewer for this comment and the text has implemented as follows, also based on Reviewer 3 comments:”… Deletions of 3p21 region, enclosing BAP1 gene are also reported in 33 MPM bioptic samples (Yoshikawa Y et al. Proc Natl Acad Sci U S A. 2016;113(47):13432-13437. doi: 10.1073/pnas.1612074113.). LOH of the entire 3p21 region has been reported, whereas many of the deletions described were not contiguous, but rather they alternated along normal DNA segments, as in chromothripsis (Tubio JM et al.Nature. 2011 ;470(7335):476-7. doi: 10.1038/470476a) …”
- In the “Genetic asset” part, ‘TAZ is a paralog …]”, the reference 28 is not accurate. Actually, absence of activating mutation for Taz is also true for Yap.
A16. We agree with this comment. The text has been revised as follows: “…TAZ is a paralog to YAP, two major effectors of HIPPO pathways. MPM is one of a few cancers (it has been demonstrated on 12 out of 14 MPM samples) that harbour mutations in Hippo pathway genes (Sekido Y et al. Cancer Res. 1995;55(6):1227-31.). Thus, it would be expected that some patients with mesothelioma would show an activating mutation in TAZ, however in vivo evidence of such mutations is lacking. Absence of activating mutation for TAZ is also true for YAP (Pulito C et al. J Exp Clin Cancer Res. 2019;38(1):349. doi: 10.1186/s13046-019-1352-3) ...”
- In the “Genetic asset” part, “A germ-line mutation in BAP1 […]” and “Interestingly, BAP1 mutations […]”, there are some specific recent articles and reviews on BAP1 more relevant than reference 5.
A17. We thank the Reviewer for this suggestion. More specific and recent references have been introduced: Walpole S et al. J Natl Cancer Inst. 2018 ;110(12):1328-1341. doi: 10.1093/jnci/djy171 and Cheung M et al Transl Lung Cancer Res. 2017 ;6(3):270-278. doi: 10.21037/tlcr.2017.05.03.
- In the “Genetic asset” part, “The HGF-receptor MET has been reported […]”, to my knowledge, amplification and mutation of MET are very unfrequent in MPM.
A18. We agree with this comment. The text has been revised accordingly, as follows: “…The HGF-receptor MET has been reported to be activated in MPM due to overexpressed (Kanteti R et al. Sci Rep. 2016 ;6:32992. doi: 10.1038/srep32992), although MET gene mutations occur in 3-16 % of cases and amplification are very infrequent in MPM (Bois MC et al. Ann Diagn Pathol. 2016;23:1-7. doi: 10.1016/j.anndiagpath.2016.04.007.)…”
- In the “In vivo biological responses” part, which looks like more as a paragraph on asbestos carcinogenesis mechanism, authors only cited reference 5 of 2015. For example, they should cite the correct reference for XRCC1 and XRCC3. Furthermore, as I mention previously, other germinal mutations are present in MPM and the recent articles should be mention. Overall, the authors should go further than the review 5.
A19. We agree with this suggestion. We implemented the text as follows, also accordingly to Reviewer 3 suggestions: “ Panou et al. demonstrated, by analyzing samples from 198 patients, that a significant proportion of them carry germline mutations in cancer susceptibility genes, as BAP1, CDKN2A, TMEM127, VHL and WT1 [Panou V.J Clin Oncol. 2018; 36(28):2863-2871. doi: 10.1200/JCO.2018.78.5204) . Moreover, it appears that mutations in two genes involved in DNA repair, XRCC1 and XRCC3, along with the GSTM1 antioxidant/detoxifying protein, increase susceptibility in this patient population, as demonstrated on more than 220 MPM samples and matched controls (Betti M et al. Mutat Res. 2011;708(1-2):11-20. doi: 10.1016/j.mrfmmm.2011.01.001 and Dianzani I et al. Mutat Res. 2006 ;599(1-2):124-34. doi: 10.1016/j.mrfmmm.2006.02.005.)
- In the “Inflammatory microenvironment” part, references 14/16 are not relevant.
A20. References 14/16 have been removed, as suggested
- In the “serology” part, a reference should be indicated for the 13-protein classifier and explanations on this classifier should be clarify.
A21. We agree with this suggestion and the text as been implemented as follows, also coherently with Reviewer 3 suggestions: “…Using the Slow Off-rate Modified Aptamers (SOMA)-scan proteomic assay, a highly sensitive candidate 13-biomarker panel was discovered and validated (on 117 MPM samples and 142 asbestos-exposed control individuals) for the detection of MPM in the asbestos-exposed population with an accuracy of 92% and detection of 88% of Stage I and II disease (Ostroff RM et al. PLoS One. 2012;7(10):e46091. doi: 10.1371/journal.pone.0046091)
- In the “serology” part, a recent article suggests that BDNF could be an interesting biomarker and should be mentioned.
A22. We really thank the Reviewer for this interesting suggestion. The following paragraph has been added in the text: “…More recently, the expression of brain-derived neurotrophic factor (BDNF), a neurotrophin, has been demonstrated in MPM. In detail high BDNF expression, at the mRNA level have been reported in tumors and at the protein level in pleural effusions (PE), thus becoming as a novel specific hallmark of MPM samples. an interesting biomarker for MPM. Notably, high BDNF gene expression and PE concentration were predictive of shorter MPM patient survival in a cohort of 79 MPM tumor samples and 26 normal pleura. Moreover, BDNF activation is implicated in the PE-induced angiogenesis: this observation has potentially strong clinical implication and supports rationale of targeting angiogenesis in MPM (Smeele P et al. Mol Cancer. 2018;17(1):148. doi: 10.1186/s12943-018-0891-0)…”
- In the “Treatment failure and further investigation” part, several references are also inadequately cited such as references 37, 38, 75.
A23. We thank the Reviewer for this comment. The section has been deeply revised also based on Reviewer 2 and Reviewer 3 comments. Particularly, more detailed and updated references have been added.
- In the “Treatment failure and further investigation” part, the authors stated “In addition, immune checkpoint inhibitors like the anti-PD1 antibody pembrolizumab and the anti-PD-L1 antibody durvalumab have been unsuccessful”. They should take into account the MAPS2 clinical trials published in Lancet Oncol. in 2019.
A24. We thank the Reviewer for this indication. The text as been implemented as follows: “…However, very recently it has been reported that anti-PD-1 nivolumab monotherapy or nivolumab plus anti-CTLA-4 ipilimumab combination therapy both showed promising activity - without unexpected toxicity - in 125 MPM patients affected by relapsing disease after first-line or second-line pemetrexed and platinum-based treatments (Scherpereel A et al. Lancet Oncol. 2019;20(2):239–253. doi:10.1016/S1470-2045(18)30765-4).
- In the “Treatment failure and further investigation”, the authors relied mainly on 1 or 2 reviews. Original articles should be mentioned and they should go further than the previous reviews.
A25. We agree with this suggestion and original articles and more updated references have been introduced.
Minor comments
- Define three keywords.
- Line 245, Reference 40 in duplicate.
- Line 277, a bracket is missing.
- Line 283 YAZ instead of YAP.
- Gene names need to be italicized.
- “Treatment failure and further investigation” part is is numbered 6.1 and there is no 6.2.
- Reference 75 was cited before 74.
Answer to minor comments: we really thank the Reviewer for the careful revision of the manuscript. All the points have been revised.
Reviewer 2 Report
Review articles focused on disease are of significant educational value, as they seek to summarize historic and current data on the subject reviewed and often provide its current and future outlooks, management, etc. This is a review article on "Malignant Pleural Mesothelioma: integrating genetic profiles and microenviroment features". Despite a significant number of reviews on malignant mesothelioma already existing, I was interested to read a review focused on genetics and the microenvironment of the disease. However, the authors failed to focus on what they prepare the reader for in their title and provided a general review on the disease, with too much information in some chapters, and a lack of information in others. My comments are as follows:
Major comments:
- The authors need to think whether this review will be focused on integrating genetic profiles and microenvironment features. Assuming yes, then I would suggest they change the text layout and content to achieve this.
- Assuming "yes" to comment 1 above, Chapters 2 (Epidemiology) and 3 (Asbestos and other causative agents) need to be smaller, they might even be combined.
- Chapter 3 is a bit problematic in its content. Paragraph 3.2 describes the the story of how SV40 was considered a cause of mesothelioma. I think this paragraph is well out of date. It has been proven and is now a consensus that SV40 did not play a role in mesothelioma causation. I would therefore find no space for such paragraph in this review. If the authors want to mention SV40 as an aetiology that was explored but was not proven, then they could do so in Paragraph 3.3. Two aetiologies that are not mentioned are: 1. genetic predisposition to mesothelioma, following chronic exposure to erionite (studies from Capadoccia, Turkey) and 2. radiation therapy. They can just be mentioned with a couple of references.
- I do not understand the need for the whole Chapter 4 being devoted to "Current Diagnostic and Staging Systems". I would make the paragraphs on Imaging and Thoracoscopy smaller and maybe intergrate a paragraph on current treatments (surgery, radiotherapy, chemotherapy) here.
- Minor comments
- Paragraphs 5.2, and 5.3 should, in my opinion, be a whole chapter with expanded information on genetic profiling. I apologize in advance if I failed to find the following important references, but if they haven't been included I suggest they are:
- Bueno R, et al. Comprehensive genomic analysis of malignant pleural mesothelioma identifies recurrent mutations, gene fusions and splicing alterations. Nat Gen, 2016
- Kiyotani K. Integrated analysis of somatic mutations and immune microenvironment in malignant pleural mesothelioma. OncoImmunology, 2017
6. Paragraph 5.4 should also be expanded to include more information on mesothelioma tumour microenvironment features, especially the immune aspect of it, such as the role of stroma, cytokines, inhibitory receptors, dendritic cells and hypofunctionality of T cells. See the references below:
-
Awad MM, et al. Cytotoxic T Cells in PD-L1-Positive Malignant Pleural Mesotheliomas Are Counterbalanced by Distinct Immunosuppressive Factors. Ca Immun Res, 2017
-
Klampatsa A, et al. Phenotypic and functional analysis of malignant mesothelioma tumor-infiltrating lymphocytes. Oncoimmunology, 2019
7. Paragraph 6.1 needs to include other immunotherapeutic strategies currently being explored, including dendritic cell-based therapies, CAR T-cell therapies etc.
Minor comments:
Line 19: pleural mesothelioma accounts for around 80% (not 90%)
Lines 21-22: a progressive rise in MPM incidence has been already observed. Please re-phrase
Line 22: I would delete "the current concept is that" and start the sentence from "The tumour..."
Line 257: mesothelioma's mutational burden is moderate, not poor
Author Response
Reply to Reviewer 2
Comments and Suggestions for Authors
Review articles focused on disease are of significant educational value, as they seek to summarize historic and current data on the subject reviewed and often provide its current and future outlooks, management, etc. This is a review article on "Malignant Pleural Mesothelioma: integrating genetic profiles and microenviroment features". Despite a significant number of reviews on malignant mesothelioma already existing, I was interested to read a review focused on genetics and the microenvironment of the disease. However, the authors failed to focus on what they prepare the reader for in their title and provided a general review on the disease, with too much information in some chapters, and a lack of information in others. My comments are as follows:
We really thank the Reviewer for careful reading of the manuscript and the constructive remarks. We have deeply revised the structure of the paper and updated the reference section. Below the point-by-point answers (A).
Major comments:
- The authors need to think whether this review will be focused on integrating genetic profiles and microenvironment features. Assuming yes, then I would suggest they change the text layout and content to achieve this.
A1. We thank the Reviewer for this fruitful comment, and we have modified the manuscript accordingly.
- Assuming "yes" to comment 1 above, Chapters 2 (Epidemiology) and 3 (Asbestos and other causative agents) need to be smaller, they might even be combined.
A2. We thank the Reviewer for this suggestion, and we have combined and shortened the two chapters, accordingly and taking under consideration the comments of the other Reviewers.
- Chapter 3 is a bit problematic in its content. Paragraph 3.2 describes the the story of how SV40 was considered a cause of mesothelioma. I think this paragraph is well out of date. It has been proven and is now a consensus that SV40 did not play a role in mesothelioma causation. I would therefore find no space for such paragraph in this review. If the authors want to mention SV40 as an aetiology that was explored but was not proven, then they could do so in Paragraph 3.3. Two aetiologies that are not mentioned are: 1. genetic predisposition to mesothelioma, following chronic exposure to erionite (studies from Capadoccia, Turkey) and 2. radiation therapy. They can just be mentioned with a couple of references.
A3. We thank the reviewer for raising this critical issue. The story of SV40 has been removed whereas the two aetiologies, namely genetic predisposition after exposure to erionite and radiation therapy, have been added. Overall, also based on Reviewer 1 and Reviewer 3 comments, the text has been modified as follows: Moreover, MPM has been associated with exposure to erionite, a zeolite mineral with some physical properties similar to asbestos which is widespread in some villages in Cappadocia (Turkey) and some areas of North America. Comparably, a cluster of deaths from pleural mesothelioma has been reported for Biancavilla (Sicily), in Italy. Subsequent studies demonstrated that those MPM cases were related to the patient exposure to fluoro-edenite, a material extracted from quarries which features morphology and composition like that of minerals of the tremolite-actinolite series… While it’s possible that these patients were unknowingly exposed, genetic analysis and other studies have led to the suspicion that chemicals such as nitrosamines, nitrosureas, potassium bromate, ferric saccharate, as well as genetic predisposition following chronic exposure to biopersistent minerals and radiation therapy are all culprits. Simian Virus 40 (SV40) infection was previously explored as aetiologic agent but it was not proven…”.
- I do not understand the need for the whole Chapter 4 being devoted to "Current Diagnostic and Staging Systems". I would make the paragraphs on Imaging and Thoracoscopy smaller and maybe intergrate a paragraph on current treatments (surgery, radiotherapy, chemotherapy) here.
A5. We thank the reviewers because this point is useful to clarify the main focus of the paper. We have shortened and integrated the section “Current Diagnostic and Staging Systems" into that focused on therapeutic approaches, also taking under consideration the suggestions from Reviewer 1 and Reviewer 3.
Minor comments
- Paragraphs 5.2, and 5.3 should, in my opinion, be a whole chapter with expanded information on genetic profiling. I apologize in advance if I failed to find the following important references, but if they haven't been included I suggest they are:
Bueno R, et al. Comprehensive genomic analysis of malignant pleural mesothelioma identifies recurrent mutations, gene fusions and splicing alterations. Nat Gen, 2016
Kiyotani K. Integrated analysis of somatic mutations and immune microenvironment in malignant pleural mesothelioma. OncoImmunology, 2017
A1. We thank the Reviewer for this careful revision and also based on the comments of Reviewer 1 and Reviewer 3 we modified and shortened the text and introduced the references indicated.
- Paragraph 5.4 should also be expanded to include more information on mesothelioma tumour microenvironment features, especially the immune aspect of it, such as the role of stroma, cytokines, inhibitory receptors, dendritic cells and hypofunctionality of T cells. See the references below:
- Awad MM, et al. Cytotoxic T Cells in PD-L1-Positive Malignant Pleural Mesotheliomas Are Counterbalanced by Distinct Immunosuppressive Factors. Ca Immun Res, 2017
- Klampatsa A, et al. Phenotypic and functional analysis of malignant mesothelioma tumor-infiltrating lymphocytes. Oncoimmunology, 2019
A2. We agree with this comment which really improve the quality of the paper. The text has been implemented as follows: “…Dissecting the properties of these inflammatory cells within tumors will provide greater insights into the immunologic mechanisms of response and resistance to immunotherapy in this disease. Awad and colleagues showed, by applying flow cytometry to characterize 43 resected MPM specimens, distinct immunologic phenotypes in PD-L1–positive tumors as compared with PD-L1–negative ones, and in sarcomatoid/biphasic tumors vs epithelioid ones. Frequencies of T cells in the 38-patient cohort were highly variable, but showed a similar differentiation status and cellular composition, including a relatively high percentage of CD4+ T cells that expressed FOXP3 (~20%). In detail they found that PD-L1–positive and sarcomatoid/biphasic tumors have a significantly greater proportion of infiltrating T cells than PD-L1– negative and epithelioid tumors, respectively. PD-L1–positive tumors also show significant increases in T-cell proliferation and activation, along with significant increases in Tregs and expression of T-cell–inhibitory markers, such as TIM-3. The work by Klampatsa et al. extended the findings by analyzing fresh tumor and blood samples of 22 MPM cases and demonstrated high levels of the inhibitory receptor TIGIT (~60%), CD39 (~20%), and CTLA4 (~25%). Overall, they found that MPM TILs were consistently hypofunctional, mainly associated with higher numbers of CD4 regulatory Tregs and with expression of TIGIT. Although the TILs showed uniformly high levels of cytokine production. The considerable immunophenotypic variability is coherent to the variable responses obtained in MPM by PD-L1 inhibitors although other factors are involved: i) the abundance of infiltrating lymphocytes; ii) co-expression of multiple inhibitory receptors on T cells; iii) the influence of MDSCs and tumor-associated macrophages…”
- Paragraph 6.1 needs to include other immunotherapeutic strategies currently being explored, including dendritic cell-based therapies, CAR T-cell therapies etc.
A3. We fully agree with this critical issue and thank the Reviewer for this suggestion. We have modified the text as follows: “…Overall MPM is characterized by a strong immunosuppressive component and this issue is implicated in the relatively low response rates to checkpoint inhibitors. On this basis, novel immunotherapeutic strategies are under investigation. In MPM patients, dendritic cells (DCs) have been shown to be reduced in numbers and in antigen-processing function compared to healthy controls and negatively affected survival outcomes. On this basis, DC vaccination represents a promising therapeutic strategy. In nine cases, DC immunotherapy using allogeneic MPM tumor lysate have been associated to enhanced frequencies of B cells and T cells in blood. The DENdritic cell Immunotherapy for Mesothelioma (DENIM) trial has been designed to evaluate the efficacy of autologous DCs loaded with allogenic tumor lysate (MesoPher) in MPM patients after first line treatment with chemotherapy MPM patients have been randomized to receive either DC therapy plus best supportive care (BSC) or BSC alone, with overall survival as primary end point. Although the trial is still ongoing, preliminary results demonstrate that DC therapy seem to be safe and effective as a maintenance treatment and promising novel treatment option. Another approach that has been investigated in MPM regards the administration of tumor antigen-targeted T cells with transduction of a chimeric antigen receptor (CAR). CARs are synthetic receptors that enhance T-cell antitumor effector function. Growing experimental and clinical evidence sustains the therapeutic role of CARs not only in the treatment of hematologic malignancies but also in solid tumor, including MPM. Interestingly, overexpression of the MET receptor can be effectively targeted using ligand-directed CAR T-cells. Indeed, MET re-targeted CAR T-cells exert in vivo anti-tumor activity against an established mesothelioma xenograft. Quite surprisingly, local delivery of targeted CAR-T seems to be highly promising. This hypothesis is allowed by the experimental observation that mesothelin-targeted CARs induce antitumor activity when subcutaneously injected in mod animal models of MPM and that intra-pleural T cell administration is vastly superior to systemic infusion in orthotopic MPM models. Another promising approach regards manipulation of induced pluripotent stem cells (iPSC) which express tumor-associated antigens vaccine promotes an antigen-specific anti-tumor T cell response. iPSC vaccines prevent tumor growth in syngeneic models of solid cancers including MPM by promoting an antigen-specific anti-tumor T cell response.
Minor comments:
Line 19: pleural mesothelioma accounts for around 80% (not 90%)
Lines 21-22: a progressive rise in MPM incidence has been already observed. Please re-phrase
Line 22: I would delete "the current concept is that" and start the sentence from "The tumour..."
Line 257: mesothelioma's mutational burden is moderate, not poor
A.We thank the Reviewer for careful revision of the text and we correct as suggested.
Reviewer 3 Report
The manuscript “Malignant Pleural Mesothelioma: integrating genetic profiles and microenvironment features” by Abbott et al recapitulates the knowledge on malignant pleural mesothelioma. The goal of this review is to describe the network between the genetics and the microenvironment. The overall concept is interesting and appropriate. However, the manuscript presents some critical flaws.
- Most of the cited studies have been performed on a small number of samples; therefore, many findings could change when repeated. the authors should indicate the number of the patients included in each study.
- Most references are old and many other manuscripts have been published on the same topic. For example, more recent genetic studies on large cohorts as well as recent findings on HMGB1 are not included in the manuscript.
- The incidence and the epidemiology of mesothelioma is reported for the Italian patients. It is not clear if the authors are characterizing the disease in relation to the Italian population.
- There are several more recent manuscripts about the molecular characterization of the EMT spectrum in mesothelioma, the authors report only the work from Reynies et al
- The authors should revise the manuscript and add references in few places. For example, for the sentence “MPM has a uniquely poor mutational landscape which appears to derive from a selective pressure operated by the environment” no reference is cited.
Minor: MPM and MM are used as abbreviations for mesothelioma, please keep the same abbreviation in the entire manuscript. There are typo errors to be fixed.
Author Response
Reply to Reviewer 3
Comments and Suggestions for Authors
The manuscript “Malignant Pleural Mesothelioma: integrating genetic profiles and microenvironment features” by Abbott et al recapitulates the knowledge on malignant pleural mesothelioma. The goal of this review is to describe the network between the genetics and the microenvironment. The overall concept is interesting and appropriate. However, the manuscript presents some critical flaws.
We thank the reviewer for the thoughtful review of our work. We have deeply revised the structure of the paper and updated the reference section. Below the point-by-point answers (A).
- Most of the cited studies have been performed on a small number of samples; therefore, many findings could change when repeated. the authors should indicate the number of the patients included in each study.
A1. We fully agree with this critical issue. The number of patients and control has been added for each study reported in the text.
- Most references are old and many other manuscripts have been published on the same topic. For example, more recent genetic studies on large cohorts as well as recent findings on HMGB1 are not included in the manuscript.
A2. We thank the Reviewer for this suggestion. More recent papers have been added, also according to Reviewer 1 and Reviewer 3 comments. Within respect to the role of HMGB1, section has been extended as follows: “…HMGB1 functions as a ‘master switch’ by which the chronic inflammation that drives mesothelioma growth is initiated and maintained. Overall HMGB1 plays a crucial role in MPM onset and progression according to the following mechanisms: i) asbestos-induced effector since its secretion by mesothelial or immune cells is highly responsive to asbestos fiber stimulation; ii) inflammatory and epithelial-to-mesenchymal transition mediator. For instance, it induces tumor necrosis factor-α secretion by macrophages thus triggering chronic peritumoral inflammation. Moreover, HMGB1 can increase the expression of cadherins thus promoting cellular mesenchymal differentiation associated to malignant phenotype. In this perspective the serum level of HMGB1 is considered to be a predictive biomarker for monitoring occupational workers and subjects at higher risk to develop MPM, although preliminary reports have been conducted in limited population. Notably, it has been reported that therapeutic levels of aspirin and its metabolite salicylic acid can suppress growth, migration, invasion, wound healing, and anchorage-independent colony formation of HMGB1-secreting human mesothelioma cells...”
- The incidence and the epidemiology of mesothelioma is reported for the Italian patients. It is not clear if the authors are characterizing the disease in relation to the Italian population.
A3. We agree with this point. Based also on the comments raised by Reviewer 1 and Reviewer 2, this point has been removed.
- There are several more recent manuscripts about the molecular characterization of the EMT spectrum in mesothelioma, the authors report only the work from Reynies et al.
A4. We thank the Reviewer for pointing out this critical issue. We implemented the text coherently: Epithelial to mesenchymal transition (EMT) results in physiological and phenotypic changes which allows epithelial cells to acquire a mesenchymal phenotype. Transforming Growth Factor β (TGF-β) plays a crucial role in promoting EMT. Indeed it has been reported in vitro that asbestos might induce EMT by downregulating the expression of epithelial markers (E-cadherin, β-catenin, and occluding), and contemporarily, by upregulating mesenchymal markers, such as fibronectin, α-SMA, and vimentin. EMT is also mediated by hypoxia inducible factor 1 (HIF- 1α) through expression of EMT transcription factors such as SNAIL, SLUG, and TWIST. In a similar fashion, by modulating cadherin activation, acts mesothelin, which expression is able to promote an EMT-associated phenotype in MPM cells. Moreover, calretinin, a Ca2+-binding protein, is implicated in inducing EMT, through the increase of focal adhesion kinase (FAK) expression and/or FAK phosphorylation in MPM cells. Calretinin (CR), through a feedback loop, negatively regulates septin 7, an essential cellular component implicated in the final steps of cell division, as a strong Bt-dependent gene regulatory protein binding to the promoter of CALB2. Thus, both CR and septin 7 represent active transducers in MPM genesis and promising actionable targets, as well. Notably, those cells show a higher resistance to cisplatin due to increased Wnt signaling.
- The authors should revise the manuscript and add references in few places. For example, for the sentence “MPM has a uniquely poor mutational landscape which appears to derive from a selective pressure operated by the environment” no reference is cited.
A5. We thank the Reviewer for this suggestion and modified the text as indicated, also coherently to Reviewer 1 and Reviewer 2 comments. In detail we introduced the following reference to the sentence above reported: Blum Y et al. Dissecting heterogeneity in malignant pleural mesothelioma through histo-molecular gradients for clinical applications. Nat Commun. 2019; 10(1):1333. doi: 10.1038/s41467-019-09307-6.
Minor: MPM and MM are used as abbreviations for mesothelioma, please keep the same abbreviation in the entire manuscript. There are typo errors to be fixed.
We thank the Reviewer for this careful revision. We correct abbreviation and typing
Round 2
Reviewer 1 Report
The authors took into account all my previous comments, I still have some minor comments.
Minor comments
In the reply to my comment 11, the authors stated « encompassing the desmoplastic subtype which percentage is largely overestimated ». I don’t think it is necessary to precise that desmoplastic subtype is overestimated as it refers to a publication that is no longer cited.
In the reply to my comment 17, the authors mentionned 3-16% of MET mutation in MPM. This mutation percentage refer to two old publications and a third one in 2011, which did not find any MET mutation. Furthermore, mutations in MET gene were not found in MPM by recent NGS studies supporting that MET mutations are really rare.
In the reply to my comment 22, the sentence “thus becoming as a novel specific hallmark of MPM samples. an interesting biomarker for MPM.” has not a correct structure.
Author Response
Reply to Reviewer 1
Comments and Suggestions for Authors
The authors took into account all my previous comments, I still have some minor comments.
We really thank the Reviewer for careful revision of our work, which is now better in terms of quality and scientific message. Below the point-by-point answers (A).
Minor comments
- In the reply to my comment 11, the authors stated « encompassing the desmoplastic subtype which percentage is largely overestimated ». I don’t think it is necessary to precise that desmoplastic subtype is overestimated as it refers to a publication that is no longer cited.
A1. The statement has been removed, according to Reviewer suggestion.
- In the reply to my comment 17, the authors mentioned 3-16% of MET mutation in MPM. This mutation percentage refer to two old publications and a third one in 2011, which did not find any MET mutation. Furthermore, mutations in MET gene were not found in MPM by recent NGS studies supporting that MET mutations are really rare.
A2. We thank the Reviewer for this comment. The text has been modified as follows: “…The HGF-receptor MET has been reported to be activated in MPM due to overexpressed, not related to the occurrence of MET genetic lesions: i) MET gene amplification are very infrequent and mutations in MET gene were not found in MPM by recent NGS studies supporting that MET mutations are really rare.”
- In the reply to my comment 22, the sentence “thus becoming as a novel specific hallmark of MPM samples. an interesting biomarker for MPM.” has not a correct structure.
A3. We really thank the Reviewer for the careful revision of the text. The sentence structure has been corrected.
Reviewer 2 Report
Thanks to the authors for revising their review.
Author Response
Reply to Reviewer 2
Comments and Suggestions for Authors
Thanks to the authors for revising their review.
We really thank the Reviewer for the kind comment and the careful review, which helped to significantly improve the manuscript.
Reviewer 3 Report
The authors have improved the manuscript following some suggestions. However, a detail molecular analysis is missing. For example (and this is just an example) at least 4 manuscripts have been published about characterization of genetic profiles and EMT, but only one is discussed in this review.
The title does not reflect what is described in the test.
Author Response
Reply to Reviewer 3
Comments and Suggestions for Authors
The authors have improved the manuscript following some suggestions. However, a detail molecular analysis is missing. For example (and this is just an example) at least 4 manuscripts have been published about characterization of genetic profiles and EMT, but only one is discussed in this review.
A1. We thank the Reviewer for this comment which aims at improving quality of the manuscript. The sections regarding EMT phenomena as well as genetic asset have been deeply revised and implemented as follows: “…In ambiguous cases, a rare transitional mesothelioma (TM) pattern may be diagnosed by conventional pathology either as epithelioid, biphasic or sarcomatoid MPM. Morphologic characteristics that favor transitional pattern include sheet-like growth of cohesive, plump, elongated epithelioid cells with well-defined cell borders and a tendency to transition into spindle cells. Detection of homozygous deletion of the CDKN2A(p16) gene compared to BAP1 loss through fluorescence in situ hybridization (FISH) on the spindle cell component could be useful to separate ambiguous cases from benign florid stromal reaction and distinguish true sarcomatoid component of biphasic MPM. Very recently, RNA sequencing unsupervised clustering analysis revealed that TM grouped together and were closer to sarcomatoid than to epithelioid MPM”…”Interestingly, long non-coding RNA (lncRNA) fragments have been shown to play diverse roles in EMT and in aggressiveness of MPM and differential signatures which could distinguish between epithelioid and sarcomatoid differentiation have been reported.”…” The molecular basis of EMT involves multiple changes in expression, distribution and/or function of transducers, including extracellular matrix and plasma membrane proteins such as periostin, vimentin, integrins, matrix metalloproteinases (MMPs) and cadherins, as well.”…”However, the exposure of MPM cells to growth factors such as FGF2 or EGF can induce a fibroblastoid morphology, associated to invasive properties, namely scattering, decreased cell adhesion and increased invasiveness.This behavior is mainly related to MAP-kinase pathway activation and quite independent of TGFβ or PI3-kinase signaling. Subsequent microarray analysis demonstrated differential expression of MMP1, ESM1, ETV4, PDL1 and BDKR2B in response to both growth factors and in epithelioid vs sarcomatoid MPM. A protein expression analysis on tissue microarray from 352 MPM samples, demonstrated that High expression of membranous EGFR, integrin β1 and nuclear p27 correlated with epithelioid differentiation whereas high expression of cytoplasmic tumoral and stromal periostin with the sarcomatoid histotype. Notably low expression of periostin in the tumour cell cytoplasm were found to be independent factors for better overall survival. Similarly, high expression of PTEN, which is known to be implicated in EMT in cancer, acts as positive prognostic factor.”…” Most alteration found affected the p53/DNA repair and phosphatidylinositol 3-kinase pathways, as well as genes involved at transcription level or expression data, such as SETDB1.”…”Within respect to gene copy number analysis, an interesting paper by Hylebos et al. analyzed an MPM-cohort (85 cases) for which genomic microarray data available through ‘The Cancer Genome Atlas’ (TCGA) and a validation cohort of 21 cases. Losses on chromosomes 1, 3, 4, 6, 9, 13 and 22 and gains on chromosomes 1, 5, 7 and 17 were found in at least 25% and 15% of MPMs, respectively. Besides the above described M-associated genes, CDKN2A, NF2 and BAP1, other interesting (and not previously described) genes carried a copy number loss ( EP300, SETD2 and PBRM1) and four cancer-associated genes showed a high frequency of amplification ( TERT, FCGR2B, CD79B and PRKAR1A). In mice combinatorial deletions of Bap1, Nf2, and Cdkn2a result in aggressive mesotheliomas, defined by stem cell-like potential. Previous analysis from the same group revealed gene rearrangements in other unexpected candidate genes. Among them the mitogen-activated protein kinase kinase 6 gene (MAP2K6), which encodes a kinase that phosphorylates p38 in response to stress and inducing apoptosis. Another interesting candidate was dipeptidyl-peptidase 10 gene (DPP10) which impacts on cell cycle regulation by binding specific voltage-gated potassium channels and modulating their function. Finally, amplification of dihydrofolate reductase gene (DHFR) and pterin-4-alpha-carbinolamine dehydratase 2 (PCBD2) an enzyme important in folate metabolism, were detected.
Whole transcriptome analysis has been used to identify differential gene expression and clustering predictive and prognostic signatures in cancer. Single nucleotide variants were firstly detected on four MPM frozen samples compared to one lung adenocarcinoma and one normal lung sample through pyrosequencing analysis. They occurred in a number of genes, namely x-ray repair complementing defective repair in Chinese gene (XRCC6), ARP1 actin-related protein 1 homolog A, centractin alpha gene (ACTR1A), ubiquinol-cytochrome c reductase core protein 1 gene (UQCRC1), proteasome 26S subunit, non-ATPase 13 gene (PSMD13), PDZK1 interaction protein 1 gene (PDZK1IP1), collagen, type V alpha 2 gene (COL5A2), and matrix remodeling associated 5 gene (MXRA5) which encode for proteins that were either previously linked to a possible role in tumorigenesis or were found to be overexpressed in different human tumors. Profiles of alternative splicing events have been also generated, such as those involving actin gamma 2 smooth muscle enteric gene (ACTG2), cyclin dependent kinase 4 (CDK4), collagen, type III, alpha 1 gene (COL3A1), and thioredoxin reductase 1 gene (TXNRD1).The most well-studied species of the non-coding transcriptome are microRNAs (miRNAs) are known to modulate gene expression in cancer. Within respect to MPM, miR-30b was found to be overexpressed in MPM and locates to 8q24, a frequently gained region in mesothelioma. Likewise, miR-34 and miR-429 located at 1p36, as well as miR-203 located at 14q32, were not expressed in tumor samples and represented regions frequently affected by DNA copy-number loss. Transcriptomic analysis has been also used to assess the differential transcriptional expression of wound-healing-associated genes in MPM during the EMT process. Overall, 30 wound-healing-related genes were significantly deregulated, among which potential targets of hsa-miR-143, hsa-miR-223, and the hsa-miR-29 miRNA family members. Out of those genes, ITGAV gene expression has been found to display prognostic value, been associated to lower overall survival. A comprehensive, multi-platform, genomic study of 74 MPM samples, as part of The Cancer Genome Atlas (TCGA) showed that poor prognosis subset showed higher aurora kinase A mRNA expression in association with upregulation of PI3K and mTOR signaling pathway. The integrative analysis allowed the identification of prognosis clusters. For instance, poor prognosis signature had a high score for EMT-associated gene expression, which was characterized by high mRNA expression of VIM, PECAM1 and TGFB1, and low miR-200 family expression. These tumors also displayed MSLN promoter methylation and consequent low mRNA expression of mesothelin, which is a marker of differentiated mesothelial cells, as reported in sarcomatoid MPM and the sarcomatoid components of biphasic MPM. Interestingly, the mRNA expression of VISTA, a negative immune checkpoint regulator primarily expressed on hematopoietic cells, was strongly inversely correlated with EMT score, being VISTA mRNA levels were highest in the epithelioid subtypes. Moreover, an unsupervised analysis of RNA-sequencing data of 284 MPMs identified a continuum of molecular profiles associated to disease prognosis. In particular, immune and vascular pathways emerged as the major sources of molecular variation, and specific profiles were detected: a hot bad-prognosis profile, with high lymphocyte infiltration and high expression of immune checkpoints and pro-angiogenic genes; a cold bad-prognosis profile, with low lymphocyte infiltration and high expression of pro-angiogenic genes; and a VEGFR2+/VISTA+ better-prognosis profile, with high expression of immune checkpoint VISTA and pro-angiogenic gene VEGFR2. It is well known that asbestos induces MPM also involving indirect effects, mainly oxidative stress associated to reactive oxygen species production and DNA-damage. These processes ultimately increase mutation rates and promote malignant transformation. ROS exposure induces methylation of the gene promoter via a specific recognition site to which DNMT1 and PARP1 are recruited, linking DNA damage and DNA methylation. Prolonged ROS exposure induces demethylation by oxidizing the 5-methycytosine to produce 5-hydroxymethylcytosine, which is catalyzed by ten-eleven translocation methylcytosine dioxygenase (TET) family of enzymes. Hypomethylation of genomic DNA is associated with genomic instability, which in combination with genetic alterations (chromosome deletion), contribute to malignant transformation. These changes entail DNA oxidation events, post-translational modifications of histones proteins, and DNA methylation. Exposure to asbestos might affect miRNAs expression through epigenetic regulation: a first example regards miR-126. Its expression increases as an adaptive response to asbestos exposure and may proceed to the loss of its expression because of DNA damage accumulation and chromosome deletion, thus leading to carcinogenesis. Interestingly, miR-103 was reported to be significantly down-regulated in the blood cell fraction of 23 patients with MPM, compared to 17 subjects formerly exposed to asbestos, and 25 healthy controls. The differential expression allowed discriminating between MPM patients and asbestos-exposed controls with a sensitivity of 83% and a specificity of 71%. Similarly, the expression of miR-625-3p was reported to be significantly higher in plasma/serum of 30 MPM patients and allowed to discriminate between cases and controls, defined as 14 healthy subjects and 10 subjects with asbestosis. More recently, MPM-specific RNA-based biomarker panels have been detected including DNA damage regulated autophagy modulator 1 (DRAM1) and arylsulfatase A (ARSA), together with their epigenetic regulators: the microRNA (miR-2053) and the lncRNA RP1-86D1.3. Overall, these circulating signatures should have important features such as low invasiveness and high specificity, which could play a critical role in next future early detection of MPM. These findings give rise to novel attention to availability of compounds that modulate epigenetic modifications, such as histone acetylation or DNA methylation in therapeutic perspective.
The title does not reflect what is described in the test.
A2. We agree with this issue and modified the title as follows: “Malignant Pleural Mesothelioma: genetic and microenviromental heterogeneity as unexpected reading frame and therapeutic challenge”
Round 3
Reviewer 3 Report
The manuscript is ready for publication.